# Transition Processes in Technological Systems: Inspiration from Processes in Biological Evolution

**DOI:** 10.3390/biomimetics10060406

**Published:** 2025-06-16

**Authors:** Martin Möller, Olga Speck, Harishankar Thekkepat, Thomas Speck

**Affiliations:** 1Cluster of Excellence livMatS @ FIT—Freiburg Center for Interactive Materials and Bioinspired Technologies, University of Freiburg, Georges-Köhler-Allee 105, 79110 Freiburg, Germany; olga.speck@biologie.uni-freiburg.de (O.S.); harishankar.thekkepat@livmats.uni-freiburg.de (H.T.); thomas.speck@biologie.uni-freiburg.de (T.S.); 2Öko-Institut Consult GmbH, Merzhauser Str. 173, 79100 Freiburg, Germany; 3Plant Biomechanics Group @ Botanic Garden Freiburg, University of Freiburg, Schänzlestr. 1, 79104 Freiburg, Germany

**Keywords:** energy transition, exnovation, fitness, generation, innovation, MLP (Multi-Level Perspective), niche, path dependency, population, selection

## Abstract

With environmental challenges intensifying, a fundamental understanding and sustainable management of ongoing transition processes are crucial. Biological evolution provides valuable lessons on how to adapt and thrive under changing conditions. By studying its key principles, we identified analogies between biological evolution and technological transitions in terms of both the Multi-Level Perceptive and the path dependency model. The comparative study also revealed that, despite contrasting time scales, the generation-based and version-based developments are comparable. In addition, interesting similarities were found in the increase and decrease of variety and between fitness and consistency. The lessons learned from biology include “Give it a try”, “Do not close for reconstruction”, and “Keep older versions in the innovation process”. Based on this comparison, we aim to gain insights for a better understanding of how to manage technology transitions and to derive concrete indicators for assessing and monitoring them. In doing so, we can provide action-oriented guidance for developing more sustainable technological solutions for major ongoing transitions, such as the energy transition.

## 1. Introduction

### 1.1. Challenges of Transition Processes

Technological systems are inherently dynamic and characterized by continuous change. Often, such change is rather subtle and characterized by incremental innovation; however, in some cases it can be truly fundamental. A prominent example of such a fundamental change is our energy system. In order to meet international climate change targets, a transition to carbon neutral electricity and heat production must be achieved within a relatively short period of time, by mid-century at the latest. Ongoing transition processes often face fundamental challenges that can impede or even prevent the change toward more sustainable production and consumption patterns. One of the main challenges is the so-called “lock-in effect” of existing socio-technical systems, which have become deeply institutionalized and resistant to change [1]. Well-established systems, such as fossil fuel-based energy infrastructures and automobile-oriented transportation networks, create path dependencies that are difficult to overcome.

In addition to environmental issues, ongoing transition processes often face profound social and economic challenges, since changes in production and consumption patterns can result in job losses and economic disruptions [2]. To maintain public support, it is therefore crucial to ensure social equity in ongoing transition processes. Moreover, the transition to a more sustainable society requires fundamental changes in consumption patterns, including a paradigm shift away from overconsumption and short-term thinking [2].

Addressing these challenges may greatly benefit from learning from nature. Over more than 3.8 billion years of biological evolution, nature has developed remarkable traits and concepts that have sustained life on Earth. A particularly compelling example is multifunctional hierarchical material systems that are fully degradable and part of natural ecosystems [3]. By studying evolutionary principles, biological concepts, and traits of organisms, as well as biological materials and their remarkable properties, opportunities arise to better understand and manage ongoing transition processes in technological systems and ultimately to develop more sustainable technologies and materials systems. In other words, these “principles of life” can serve as design lessons and benchmarks for human innovation. By applying and operationalizing nature’s role models, we can develop more effective strategies for targeted transition processes and address complex sustainability challenges across multiple disciplines [4].

### 1.2. Sustainability and Sustainability Assessment

Sustainability is a complex and multifaceted term. In the “Brundtland Report” issued by the independent expert commission established by the United Nations in 1983, sustainable development is defined as “development that meets the needs of the present without compromising the ability of future generations to meet their own needs” [5]. The three dimensions of sustainability—environmental, economic, and social—are defined as interdependent and must be balanced over the long term. Furthermore, the Brundtland Report defines sustainable development as a necessary transition process for both the economy and society.

“Yet in the end, sustainable development is not a fixed state of harmony, but rather a process of change in which the exploitation of resources, the direction of investments, the orientation of technological development, and institutional change are made consistent with future as well as present needs”. [5](the source)

As an anthropocentric approach, sustainability has the character of a normative guiding principle. Hence, an assessment of the sustainability of technologies or materials systems is ultimately normative, whereby several different normative reference systems have been defined for the assessment step. In the past, the lack of standardized metrics, indicators, and reporting frameworks has made it difficult to accurately measure and communicate progress toward sustainability. With the adoption of the 2030 Agenda and its Sustainable Development Goals (SDGs) in 2015, all 193 member states of the United Nations agreed for the first time on a universal set of binding and time-bound normative goals. These 17 SDGs, along with the associated 169 targets and 231 indicators, increase comprehensibility and the obligation to understand sustainability. Therefore, the SDGs can be considered the currently valid normative basis for a sustainability assessment [6].

A new tiered framework, TAPAS (Tiered Approach for Prospective Assessment of Benefits and Challenges), has been developed based on the SDGs to evaluate the sustainability of emerging technologies and materials [7]. TAPAS enables innovation actors to conduct robust assessments autonomously and as early as possible to identify risks and utilize existing opportunities. The framework incorporates the three core sustainability strategies of efficiency, consistency, and sufficiency to promote more sustainable production and consumption patterns [7,8].

Although sustainability is an anthropocentric and teleological approach that does not exist in biology, it is possible to attribute biological concepts to the three sustainability strategies in order to make them more concrete. Using the growth form of lianas as an example and analyzing its fundamental biological concepts, it has been demonstrated how we can deepen our understanding and refine the definitions of sustainability strategies to achieve a more comprehensive and robust sustainability assessment. Moreover, biological concepts can serve as a source of inspiration for more sustainable solutions within the technosphere [8,9].

### 1.3. Background Information About Biological Evolution

Over 3.8 billion years of biological evolution, a vast diversity of organisms has evolved, including fossil and recent plants, animals, fungi and bacteria. Environmental constraints act as evolutionary pressures that modify the traits of organisms and generate adaptations that enhance their evolutionary fitness. In biology, adaptations are understood as hereditary genetic changes in populations over evolutionary time [10]. A prime example is fire-adapted plants (phyrophytes) that are able to survive exposure to heat in various fire landscapes on Earth. Pyrophytes protect their living tissue with thick bark and benefit from fire events either directly (e.g., fire initiates cone opening and seed release), or indirectly (e.g., fewer competing plants of fire-sensitive species remain, seeds germinate in the ash-fertilized soil) [11,12]. Another prime example is drought-adapted plants (xerophytes), which exhibit sophisticated water management adaptations and can survive in semi-arid and arid landscapes [13].

Biological evolution is characterized by random variation in the genetic information of individuals (genotype) and populations (gene pool), and the natural selection of successive generations based on observable traits (phenotype) including morphological, developmental, biochemical, and physiological properties. The key principles of biological evolution include the random increase in variability (e.g., mutation [14], recombination), the random decrease in variability (e.g., isolation, genetic drift [15]), and natural selection. Mutation refers to small random changes in genetic information, recombination involves the random rearrangement of parental genetic material, genetic drift is the change in frequency of an existing gene variant (allele) in a population by chance, and isolation is the interruption of gene flow between populations. The theory of evolution by natural selection was described in detail by Charles Darwin in his book *On the Origin of Species* published in 1859 [16]. Darwin’s focus was not on the process of speciation but rather on the gradual change of individual species over time through natural selection, defined as the varying reproductive and survival success of offspring best suited to their natural environment [17].

We can observe short-term changes in the breeding of agricultural animals and plants with targeted artificial selection. Long-term changes, such as the evolution or extinction of species, may occur over millions of years of natural selection and can be reconstructed from fossils and recent organisms [17]. Several hypotheses have been proposed regarding the driving force behind the phenotypic evolution of multicellular body plans [18], whether by mutation or natural selection. However, in this article, we will not contribute to the selectionism–mutationism controversy [14]. Instead, we aim to explain the principles of biological evolution in a simplified way as inspiration for a possible application to the challenge of technology transition.

According to von Gleich et al. [19], there are three levels of learning from biological models for technological applications: (i) learning from the results of evolution, such as the reversible adhesion of plant hooks in animal fur as a model for the hoop-and-loop fastener (velcro) [20], (ii) learning from the success principles of evolution, such as biological concepts of the growth form liana as source of inspiration for the three major sustainability strategies of efficiency, consistency, and sufficiency [9], and (iii) learning from evolutionary principles of the evolution process, such as the Evolution Strategy, a biomimetic optimization method. In the mid-1960s and early 1970s, Ingo Rechenberg [21] and Hans-Paul Schwefel [22] developed the “Evolution Strategies”, abstracted from the functional principles of mutation, recombination, isolation, and selection of Darwinian evolution [23,24,25].

In recent years, various attempts have been made to compare the process-based principles of biological evolution with technological development. Arthur [26,27] coined the term “combinatorial evolution”, meaning that all technologies, like all species, can be traced back to earlier technologies. However, radically novel technologies do not emerge through the Darwinian mechanism of cumulative small changes and natural selection; rather, they emerge from the combination or integration of earlier technologies [28]. Solée et al. [29] describe the history of technology as a parallel, human-driven experiment to biological evolution. Although artifacts—unlike organisms—cannot self-reproduce, they share population-driven dynamics, cost constraints, and the diversification, convergence, and extinction of properties. The authors also emphasize that technological innovations are highly dependent on the combination of pre-existing elements, as illustrated by the evolutionary tree of cornets.

### 1.4. Aim of the Project and Key Results

In this article, we compare biological evolution with technological transition. We aim to highlight the similarities, commonalities, and differences between these two processes and endeavor to learn from the lessons of biology for technology. Our introduction (Section 1) outlines the challenges of technological transitions within the context of sustainable development and provides some general background information on biological evolution. In Section 2, we introduce the key principles of biological evolution and present the biomimetic optimization method “Evolution Strategy”. In Section 3, we relate the technological transition process and its sustainability assessment to biological evolution, drawing analogies using the Multi-Level Perspective and path dependency model. Based on the comparison between biological evolution and technological transition, we derive starting points for metrics and indicators to improve sustainability assessments (Section 4). Section 5 presents a discussion of our findings, while Section 6 summarizes the general conclusions and provides an outlook for future work. For ease of reference, we have also included a glossary of biological and technical terms (Section 7). 

Specifically, our aim is to provide answers to the following key questions:


1.What can we learn from biological evolution in order to better understand and manage the sustainability transition processes in technology?2.How can biological evolution inspire the generation of metrics and indicators to assess the sustainability of materials systems and technologies in the context of ongoing transition processes?


We summarize our main contributions to the comparison between biological evolution and technological transition as follows:


We adopt a multilevel perspective on biological evolution, drawing an analogy to the multilevel perspective on socio-technical transitions.We apply the model of path dependency from technological transitions to biological evolution.We identify key lessons from biological evolution for sustainability transitions, including concepts such as “give random changes a try” and “keep older versions in reserve”.We derive several metrics and indicators for assessing and managing sustainability transitions, inspired by principles of biological evolution.


## 2. Inspiration from the Overall Process and Key Principles of Biological Evolution

### 2.1. Biological Evolution

In biology, evolution refers to the change in the genetic material of organisms, and thus of populations, which accumulate over generations on an evolutionary time scale. The classical theory of evolution of Charles Darwin [16] and Alfred Russel Wallace [30] is constantly expanding through new scientific findings that include paleobotany, botany, zoology, genetics, and systematic biology. Building on this combined knowledge, we now refer to the “modern synthesis”, a term coined by Julian Huxley in 1942 [31]. Before exploring this in depth, we present some key statements about how biological evolution differs from technical development processes.


**Biological evolution is a continuous process based on heredity.**
Even when new traits arise through variation in genetic material during evolution, these adaptations are built on preexisting traits, as the inheritance process provides long-term memory across the evolutionary timescale [32]. In their lectures, German zoologist Professor G. Osche vividly described the difference between biological evolution and technical developments that we translated as follows “Nature cannot display a sign saying: closed for reconstruction” [33]. This quote highlights that biological organisms must remain “functional” throughout every stage of evolution, carrying with them an “evolutionary burden” inherited from their ancestors. In contrast, technological progress can involve disruptive shifts, where entirely new technologies may emerge without relying on existing ones. A well-known example is the rise of digital technologies, which did not evolve directly from analog predecessors.
**Biological evolution is neither anthropocentric nor teleological.**
In technology, the purpose of an application is defined by engineers from the onset and is optimized throughout the development process. In contrast, biological evolution operates through a process of “trial and error”. The random increase or decrease in genetic variability can be considered as an experiment or “trial”, and the decrease in individual reproductive success (=fitness) can be interpreted as an “error”. In an interview, Professor J. Gadau of the University of Münster, Germany, aptly summarized this concept that we translated as follows: “Evolution is blind, but not random”. [34].
**Phenotypic convergence is commonly associated with adaptive evolution.**
Regardless of the respective body plan, species from independent lineages can produce the same or a very similar phenotype when they form a similar ecological niche and adapt in comparable ways to similar selective environmental constraints. Prime examples of such a convergent evolution, which have evolved in various unrelated plant lineages, are the growth forms of climbing lianas and succulents. Climbing lianas show convergent adaption in stem mechanics, water conduction, and attachment systems [35,36]. Succulents that have evolved in various arid environments on Earth [37] exhibit adaptations for maximized water uptake, minimized water loss, and optimized water storage [13].
**Adaptive radiation is commonly associated with biodiversity.**
A classic example of adaptive radiation is Darwin’s finches (cf. Section 2.1.4), which are not true finches but belong to the subfamily Geospizinae. The great diversity of these “finch” species arose from a single species that accidentally reached the Galapagos Islands. In general, islands open up new ecological niches for founder populations, resulting in the evolution of phenotypic adaptation and speciation. Another example of adaptive radiation can be found in the succulent plant genus *Aeonium* on the Canary Islands [38,39,40,41].

#### 2.1.1. Gene Pool of a Population

A population is defined as a group of individuals of the same species that live in the same habitat at the same time and are capable of reproduction [17]. Each individual in the population exhibits traits (phenotype) as a result of its genetic material (genotype). The gene pool refers to the totality of all gene variants (alleles) in a population at a given point in time. In diploid organisms, such as humans and many plants and animals, each individual has two alleles, one from the father and one from the mother. However, multiple alleles are possible in a population, and they often show a hierarchy of dominance [17]. Allele frequency describes the relative proportion of a particular allele in a specific population. Changes in allele frequency can be used to measure increases or decreases in the genetic variability of a population [17]. Figure 1a depicts a schematic drawing of various genotypes of individuals in a population that exhibit a variety of phenotypes with respect to various flower colors. In our example, the blue allele dominates the other alleles and produces blue flowers (Figure 1a).

#### 2.1.2. Increase of Genetic Variability

Genetic variability is increased through mutation and recombination. Mutation refers to small, random changes in genetic information (Figure 1b) by changing one allele, mutations can then be passed from the parental generation to the offspring [17]. Most mutations have either no effect on the organism (neutral mutations) or a negative effect (deleterious mutations). However, some can have a suddenly beneficial effect under changed environmental conditions (cf. Section 2.1.5). Figure 1b shows the genotype of a population after a mutation has occurred, resulting in an additional yellow allele and thus an additional yellow flower color. Recombination represents the random rearrangement of parental genetic material during sexual reproduction [17]. Figure 1c shows that recombination can be considered an exchange of alleles. A recombination of alleles has taken place in our example, as shown in Figure 1c, resulting in a new allele combination, namely red and yellow, and an individual with the newly emerging magenta flower color.

#### 2.1.3. Decrease of Genetic Variability

Reproductive isolation is the interruption of gene flow between populations that is essential for the process of speciation (Figure 2). The mechanisms of reproductive isolation can be caused by various types of barriers, such as geographic isolation of populations on two separate islands, temporal isolation of animal populations with different mating times, ecological isolation of populations optimally adapted to different ecological niches in the same area, or mechanical isolation in plants with flower shapes adapted only to specific pollinating insects.

Genetic drift (Figure 3) is a random change in the allele frequency of the gene pool comparing the original population and the separated smaller populations [17]. Since a random change in gene frequency is statistically more significant in smaller populations, genetic drift is an important factor in the evolution of new species and thus in speciation [15].

**Figure 1 biomimetics-10-00406-f001:**
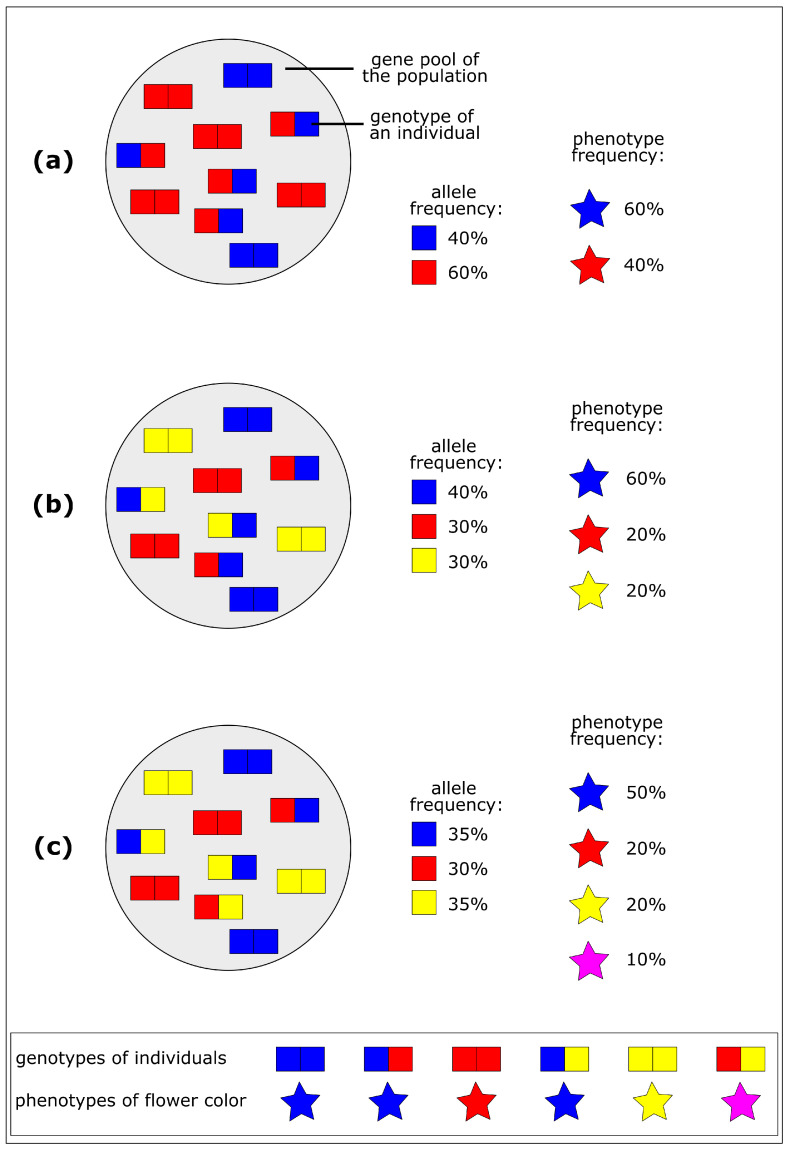
Overall processes of biological evolution resulting in increased variability. (**a**) Gene pool of a population with 10 individuals. The genotype is dependent on two alleles. In this example, the genotypes result in various phenotypes of flower color (blue and red), with the blue allele being dominant and red allele recessive. (**b**) Mutation results in a third type of allele and an additional phenotype (yellow flower color), with yellow being recessive. (**c**) Recombination results in new combinations of alleles (red and yellow) and thus in an additional phenotype (magenta flower color), with red and yellow being intermediate.

**Figure 2 biomimetics-10-00406-f002:**
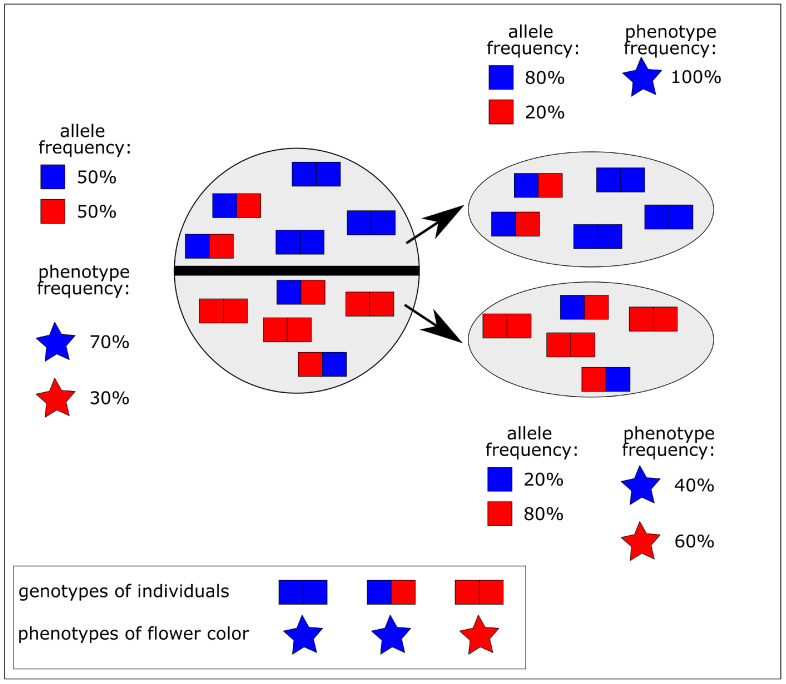
Overall processes of biological evolution resulting in decreased variability. Reproductive isolation in a population exhibiting blue and red flowers results in two populations and a change in allele and phenotype frequency. One population (shown above) has only blue flowers, while the other population (shown below) has red and blue flowers.

The bottleneck effect (Figure 3a) is a specific type of genetic drift where the population size is greatly reduced by a random event such as natural disasters like floods, wildfires, volcanic eruptions, or storms. The reduction in population size is accompanied by a reduction in genetic variability and thus in phenotypic variability, making future adaptive changes less likely [17]. In Figure 3a, the original population has blue, red, yellow and magenta-colored flowers, while caused by the bottleneck effect only red, yellow and magenta-colored flowers are left in the new population.

The founder effect is a special case of population bottleneck, where the original population is split into several small founder populations. In these smaller founder populations, some alleles may prevail while others may be completely absent [17]. Under extreme environmental conditions, however, reduced genetic variability may result in reduced chances of survival. On the other hand, reduced genetic variability may result in an offspring population that is markedly different from the parent generation. In Figure 3b, the flower colors of the original population are blue, red, yellow, and magenta, while the upper founder population has only yellow and magenta flowers, the middle population only blue and red flowers, and the lower population only blue flowers.

**Figure 3 biomimetics-10-00406-f003:**
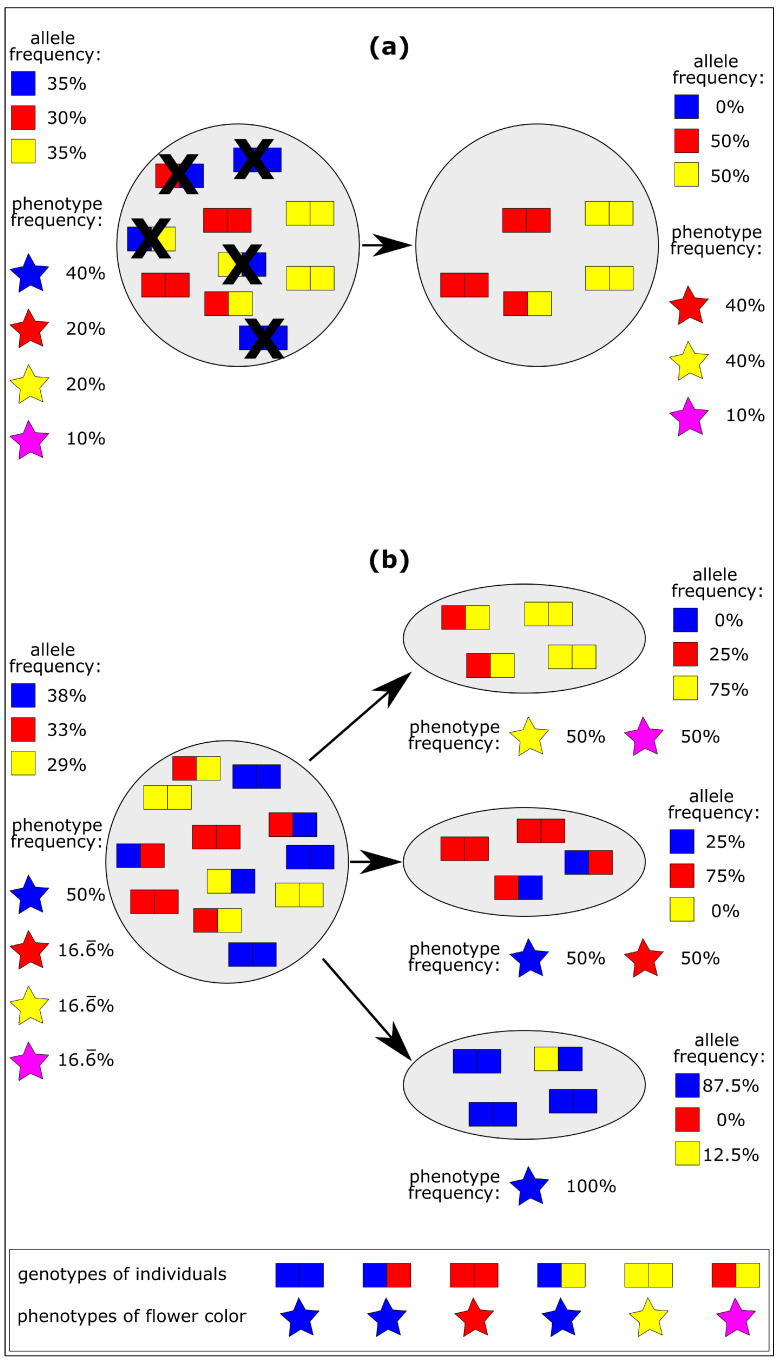
Genetic drift results in decreased variability. (**a**) Bottleneck effect caused by the random extinction of individuals by natural disasters (indicated by cross marks). (**b**) Founder effect, which results from the establishment of new populations based on the splitting off of a small number of individuals from the original population.

#### 2.1.4. Adaptive Radiation and Ecological Niches

Founder populations, on the other hand, can give rise to new species through adaptive radiation, form different ecological niches, and thus support the evolution of biodiversity [39,40]. In biology, the ecological niche is not a space that is occupied, but describes the set of relationships between a species and its environment, accounting for both biotic factors (other organisms, e.g., food, competitors, enemies, symbionts, parasites) and abiotic factors (physical factors such as temperature, humidity, water availability, salinity, and others). As outlined above, examples include Darwin’s finches on the Galapagos Islands [17] and the genus *Aeonium* (Crassulaceae) on the Canary Islands [38,39,40,41].

#### 2.1.5. Natural Selection and Biological Fitness

According to Charles Darwin, differential reproductive and survival success represent the evolutionary principle of natural selection [16]. Although frequently misinterpreted in the past, natural selection is considered to be the real controlling element of evolution since it compensates for the initial random processes of mutation and recombination, and determines the direction in which the gene pool of a population changes. The confusion over the meaning of natural selection was partly caused by the British sociologist Herbert Spencer, who introduced the term “survival of the fittest” as part of his socio-economic theories in his book *Principles of Biology* [42]. Darwin then included the term in the fifth edition of *On The Origin of Species* [16]. Although Darwin’s understanding of fitness referred to individuals who are better adapted to their immediate and local environment, today, “fitness” is often misinterpreted as “best physical condition”.

Although natural selection acts on individuals, it is populations that evolve, not the individuals themselves. Thus, since evolution is defined as a change in the genetic composition of a population from one generation to the next, natural selection acts only indirectly on the genotype. The direct effect of evolution also does not act on a single phenotypic trait, but always on the whole individual and its totality of phenotypic traits. The reproductive contribution of an organism, and thus its overall phenotype in subsequent generations, relative to the contribution of other organisms in its population is called its biological fitness or Darwinian fitness, often simply called “fitness” [17].

In 1968, the Japanese scientist Motoo Kimura formulated the neutral theory of molecular evolution [43], which states that the vast majority of variants observed in most populations are neutral to selection. For this reason, neutral mutations accumulate through gene drift rather than through selection [17]. However, if the environmental conditions and thus the abiotic and biotic selection factors change, a neutral mutation may turn out to be beneficial or deleterious.

As early as 1859, Charles Darwin already mentioned evolutionary interactions between flowering plants and insects in his book *On the Origin of Species* [16]. This phenomenon, now known as co-evolution, describes how two closely interacting species exert selective pressure on each other over evolutionary time, eventually resulting in co-adaptations [44]. A well-known example of such a mutualistic relationship is the evolution of red flower coloration in plants and the corresponding adaptation of birds as their primary pollinators (cf. changes in flower color, as shown in Figure 1, Figure 2 and Figure 3).

### 2.2. Evolution Strategy—A Biomimetic Optimization Method

The development of the biomimetic optimization method “Evolution Strategy” shows that it is possible to transfer the key principles of biological evolution to the technosphere. In the mid-1960s and early 1970s, Rechenberg [45] and Schwefel [46] developed the “Evolution Strategy”, which has been further elaborated in recent years. The Evolution Strategy is a highly robust method, meaning that even in cases of faults, errors, or mistakes, the algorithms eventually lead to an optimal solution (cf. Figure 7 published in [47]).

All evolution-inspired optimization methods share the common principle that a collection of solutions (=population) is subjected to simulated biological evolution. By randomly changing the parameter values of individuals (=biological mutation) and by recombining the parameter values of different individuals (=biological recombination), new solutions are generated (=increase of variability), which are then evaluated for their efficiency. The less efficient solutions are weeded out (=natural selection), leaving only the best-fit solutions. These individuals then form the basis for further evolutionary cycles (=generations) [25]. Table 1 provides an overview of the comparison between the principles of biological evolution and Evolution Strategy.

Evolution Strategy is a biomimetic optimization method that uses objective criteria (e.g., calculations, experimental results) or subjective criteria (e.g., taste of coffee [48]) for selection. Depending on whether this is a maximization or minimization problem, we aim to find either the highest peak or the deepest valley in the quality landscape range.

While finding the global optimum is always desirable, it is theoretically verifiable only in relatively simple tasks. Therefore, in practice, it is often more reasonable not to search for a global optimum, but to accept a local optimum that is only slightly worse than the one that will be achieved after investing even more effort in terms of money or time in further optimization [49]. The algorithm includes when to stop the optimization, for example, after a certain number of generations, after a certain period of time, or when there is no significant difference in the measured variable after *n* generations [46,47].

Dujardin et al. [50] used Switzerland as a case study to show how the large-scale transition to renewable energy can be optimized. Evolution Strategy enables the simultaneous optimization of placement and share of the renewable electricity generation technologies added to a system. The new electricity generation is combined with the existing generation as efficiently as possible while taking into account the restrictions of the electrical grid.

## 3. Transition Processes in Technological Systems

Technological transitions are deep, systemic changes that fundamentally reshape the functioning of socio-technical systems, including aspects of people, infrastructure, technology, culture, processes, procedures, goals, as well as indicators and metrics. Transitions in technology often emerge from technological innovations that gradually change [51] established systems, driving shifts in social, economic, and regulatory structures [52,53]. “Transition” involves a phased, structured shift within systems, such as energy or mobility, focusing on how these shifts happen and emphasizing the interactions between social, technological, and institutional elements.

The term “transformation” is another important term in the same discourse that refers to more fundamental, pervasive shifts that impact entire societies [54]. In the energy sector, “transition” better captures the organized shift from fossil fuels to sustainable alternatives. This term reflects the changes driven by specific innovations required to move the energy sector toward sustainability, while balancing technological and societal interactions [54].

In this context, it is essential to distinguish between technological system shifts and socio-technical system shifts. A technological system shift primarily refers to changes within the technical composition of a system, such as the introduction, replacement or upgrading of specific artifacts. These changes focus on aspects like performance and efficiency and primarily address the technical domain, with limited consideration of broader societal factors. A socio-technical system shift, in contrast, is a configuration of intuitions, actors and technologies that together fulfill societal functions. Therefore, a socio-technical shift involves a reconfiguration of both the technical and social aspects of a system, which includes a wide array of changes, such as cultural practices and regulatory frameworks, in addition to technological innovations [55]. The Multi-Level Perspective (MLP) framework is a common approach for analyzing such transitions [56,57].

### 3.1. Multi-Level Perspective in Socio-Technical Transitions

Technological transitions are processes that involve interactions between multiple levels. They involve a shift from one stable technological configuration to another, typically driven by a combination of innovation, societal, and institutional demands [52,58]. The Multi-Level Perspective (MLP) [57] is a key framework for understanding such transitions, and categorizing them into the three analytical levels: landscape, regime, and niches.

1.**Landscape** in the MLP framework represents an environment in the socio-technical system [52]. It includes global events, developments, and trends such as climate change since the 1990s, energy crises in the 1970s and 2020s, the 2008 banking crisis, and technological changes (e.g., digitization and artificial intelligence). The landscape puts pressure on the existing regime for changes [59]. For example, climate change concerns are pushing for decarbonization and the energy crisis has accelerated investment in renewable energy projects.2.**Regime** in the MLP framework represents the dominant configuration of actors, structures, rules, and norms that stabilize the existing socio-technical systems. This includes existing technologies and user practices that reinforce the status quo. The regime is usually resistant to change because it is supported by incumbent actors who benefit from the current system [52,59]. For example, in the energy regime, the dominance of fossil fuel-based energy production is reinforced by policies, infrastructure, and market preferences.3.**Niches** in the MLP framework represent spaces for innovation, where new technologies, business models, and practices emerge. Niches refer to specialized market segments or application areas in which a particular technology is uniquely suited to meet specific needs or solve distinct problems. Over time, when niche innovations align with landscape pressures and gain momentum, they challenge and eventually disrupt the existing regime [52,59]. For example, renewable electricity generation was once a niche innovation supported by government incentives and research funding.

### 3.2. Interaction Between Landscape, Regime, and Niche

Figure 4 illustrates the interactions between landscape, regime, and niches in the MLP framework, using the energy sector as an example and focusing on technological aspects. Initially, the energy regime is stable and primarily reliant on fossil fuels, represented in dark orange in the pie chart. Societal developments, such as energy crises and climate change, exert pressure on the regime, leading to instability. This opens a window of opportunity for niche innovations to emerge. Niche innovations, driven by both landscape and regime pressures, begin to affect the regime and eventually drive its reconfiguration. For example, the energy regime once dominated by fossil fuels has reconfigured into one led by renewable energy sources, shown in light orange in the pie chart.

Niche innovations gradually expand throughout technological transitions, challenging and destabilizing the regime through external pressures from the broader environment (i.e, the landscape). This process can ultimately lead to the creation of a new regime [60]. The characteristics of this paradigm shift can vary significantly depending on factors such as the nature of the technology, societal acceptance, regulatory responses, and the prevailing economic conditions [61]. The landscape also feeds back into the system, exerting pressure on niches to accelerate innovation and steer development in particular directions. As the different levels of ongoing transitions interact, they can either accelerate or hinder the transition speed, shaping the future landscape in unpredictable ways. Rapid advancements or sudden policy changes can accelerate transitions, while resistance to change can prolong or delay them. As a result, paradigm shifts may unfold faster or more slowly than anticipated.

### 3.3. Path Dependency and Lock-Ins

Path dependency refers to a phenomenon that describes how historical decisions and established practices shape the trajectory of future developments, often leading to a reliance on specific technologies or systems, hindering innovation and ongoing transitions. Since this phenomenon can be a major obstacle to sustainability transitions, examining path dependency is crucial for identifying existing triggers for change and designing policies that enable radical change and flexibility in most parts of the evolving transition [62,63]. Lock-in refers to a situation where a system becomes “snapped into place”. In such cases, alternative degrees of freedom of the further path are reduced to zero, and thus the path is clearly defined and consequently fixed to a firmly determined development [64]. Solée et al. [29] provide mathematical terms that describe this phenomenon.

An emblematic example of detrimental path dependency is the lock-in of fossil fuel infrastructure, particularly in the form of gasoline-based mobility. Path dependency can manifest itself in several ways, such as through infrastructure lock-in, where the existing network of fossil fuel infrastructure, including pipelines, refineries, and gas stations, creates a self-reinforcing system that favors the continued use of these resources [65].

Figure 5 illustrates the model of path dependency including the phases of path creation, path shaping, path dependence after a lock-in event, path breaking after an un-locking event and a subsequent restart of the cycle [64]. Conceptually, the model is a heuristic helix that represents the range of available options as a function of time.

**Path creation:** In phase I, path creation starts in a complete open space of possibilities. However, based on history and context, the options against the gray background are more likely to be pursued than others. In addition, the time component of path creation is also a challenge. On the one hand, generating sufficient momentum and identifying a critical event that catalyzes path creation is vital; on the other, maintaining persistence is necessary until stabilizing mechanisms take effect [64].**Path shaping:** In phase II, a specific path begins to emerge as certain options are selected while others are abandoned. This process is shaped by critical junctures, where each new decision is made from a constrained set of available options, all of which are influenced by previous choices [64]. Whether, when, and to what extent such an event will occur cannot be determined ex ante, but is random [66].**Path dependency:** In phase III, the so-called **lock-in** occurs [64]. Lock-ins refer to stages in systems where certain technologies, practices, or policies become deeply entrenched. At this point, established organizations and regulatory frameworks may resist change due to vested interests [67]. Understanding the reasons for lock-ins and how to avoid them is crucial because it enables actors to identify barriers to systemic change to establish more sustainable systemic practices [68].**Path breaking and restart:** In phase IV, a lock-in-break, also known as **un-locking**, occurs. However, such a disruption can only be triggered by external impulses, as internal processes tend to be self-reinforcing and resistant to change. When considering a path shift, the potential benefits must be weighed against the switching costs, which may include financial investments, time, and the effort required for learning and adaptation. If a path is successfully broken, the cycle begins anew with the creation of a new path [64].

### 3.4. Technological Transition in the Energy Sector

The energy transition is a global challenge. From 2000 to 2023, the share of renewables in global electricity generation increased from 19% to over 30%, mainly driven by solar and wind. In 2023, China was the main contributor, accounting for 51% of the additional global solar generation and 60% of new global wind generation [69], followed by the US, India, and Germany [70]. As a result of electricity generation from low-carbon sources such as wind, solar, and nuclear power, the greenhouse gas (CO_2_) intensity of global power generation has reached a new record low, 12% lower than its peak in 2007 [69].

The energy transition is an exemplary case study illustrating sustainability transitions through the MLP framework due to its interplay between various socio-technical systems. As explained, the MLP put forward that the transition occurs through interactions on three levels: the niche, the regime, and the landscape (cf. Figure 4). In the context of the energy transition, niche innovations, such as renewable energy technologies and decentralized energy systems, actively challenge the dominant fossil fuel regime, thereby driving shifts in policies, market dynamics, and consumer behavior [71]. The energy transition is further influenced by landscape pressures, including climate change and resource depletion, which create a pressing need for sustainable practices.

#### 3.4.1. Case Study: Energy Transition in Germany

The technological transition in Germany’s energy sector refers to the shift from a fossil fuel–based system toward the adoption of more sustainable energy technologies. The transition is being driven by growing societal demands for cleaner energy, along with more supportive policy frameworks and evolving market dynamics [72]. Currently, the energy transition largely involves the integration of renewable energy sources such as solar and wind, along with other technologies such as energy storage, smart grids, and demand response systems to create energy systems that have improved efficiency, consistency, and sufficiency.

Germany currently aims to achieve net-zero emissions by 2045. Phasing out CO_2_ emissions in the energy sector is a key part of achieving this target. The transition in the energy sector, referred to as “Energiewende” [73], includes phasing out nuclear energy and expanding renewable energy sources such as wind and solar power [74].

The analytical levels of the MLP framework are used to describe the key determinants of the energy transition in Germany:**The role of niche innovations**: The niche technologies of the energy transition in Germany are comprised of renewable energy systems and energy storage solutions. Solar and wind energy technologies, initially developed in niche markets over recent decades, were initially supported by policies like the feed-in tariff system, which offered remuneration to renewable energy producers. The continued scaling and efficiency improvements of renewable energy technologies have been reinforced by policies [73]. In recent years, green hydrogen has emerged as a key niche innovation, particularly important for decarbonizing hard-to-electrify sectors such as heavy industry and transportation. Germany is currently investing heavily in the development of green hydrogen and considers it a key component of its energy transition program [75]. Decentralized energy production represents another growing and increasingly popular area of innovation. Initiatives like balcony power plants allow consumers to generate their own electricity at home, with excess electricity contributing to local energy supplies. By promoting such solutions, Germany is developing a robust energy system with less dependency on centralized power generation [73].**Destabilizing the energy regime**: Historically, Germany’s energy regime was dominated by coal, gas, and nuclear power. However, more recent niche innovations such as solar, wind, and green hydrogen technologies have played pivotal roles in destabilizing the existing energy regime, supported by external landscape pressures such as global climate commitments. Coordinated political, technological, and societal efforts have accelerated this shift, leading to renewables (solar, wind, biomass, and hydropower) accounting for 59.4% of gross electricity generation in 2024 [76]. Germany’s phase-out of nuclear power in 2023 was a major milestone in the country’s energy transition, marking an end to the reliance on nuclear energy and reinforcing the commitment to expanding renewable capacity. The nuclear phase-out was largely driven by landscape pressures [73]. Nevertheless, resistance from incumbent fossil fuel industries remains a significant barrier. These industries, benefiting from decades of infrastructure and market dominance, resist transitions that could threaten their economic viability [77]. However, public pressure, global climate agreements, and technological advancements in renewable energy are gradually eroding their influence.**Pressures from the landscape**: The energy crisis triggered by the 2022 war in Ukraine has intensified pressures on the energy regime to reduce dependency on Russian gas and accelerate the adoption of renewable and alternative energy sources [78,79]. Earlier, the 2011 Fukushima nuclear accident significantly heightened global concerns about nuclear safety, prompting Germany to accelerate its nuclear phase-out [80]. These events acted as powerful landscape pressures, disrupting the stability of the existing energy regime and creating a window of opportunity for niche innovations, such as wind and solar energy, to gain momentum in driving the broader energy transition.**Building a new regime and landscape**: The future of Germany’s energy transition lies in constructing a new technological regime centered on digitization, expanded renewable energy capacity, and improved energy efficiency. Investments in smart grids and energy-efficient urban planning are changing how energy is both generated and consumed [73]. Here, consumers are becoming “prosumers”. Namely, they participate in the energy generation by generating their own electricity through systems like rooftop solar panels and feeding surplus power back into the grid [81]. This shift from consumer to prosumer marks a transition from centralized to decentralized energy systems, a distinctive characteristic of the new energy regime. Digitization plays a key role in enabling decentralized energy production and management, as well as in integrating variable renewable sources into the grid efficiently. Looking ahead, the continued development of a regime based on decentralization, digitization, and renewables will continue to fundamentally reshape Germany’s energy landscape.**Influence on social, economic, and regulatory structures**: Technological transitions, as illustrated in Germany’s energy sector, are not only about technological advancement, but also involve comprehensive changes in various other structures. For example, the transition to renewable energy sources is fostering greater community engagement and awareness of sustainability, reshaping public attitudes towards energy consumption and conservation. Economically, the transition is creating new job opportunities in novel energy sectors, while potentially displacing jobs in traditional ones. At the regulatory level, existing governance frameworks must adapt to support renewable integration, requiring updated policies that facilitate infrastructure development, regulate emissions, and incentivize the adoption of environmentally friendly technologies. The complexity of these interrelated social, economic, and regulatory structures highlights that energy transitions aimed at achieving sustainability goals are multi-dimensional processes [82].

#### 3.4.2. Gap Analysis of Technological Transition in the Energy Sector

To ensure a reliable energy supply, it is essential to identify and implement state-of-the-art systems and innovative material concepts at an early stage. Key technologies include enhanced energy storage solutions, such as next-generation batteries and grid-scale storage systems, which will assist in managing the intermittency of solar and wind generation. In parallel, the development of smart grid technologies will enable real-time energy management, enhancing supply–demand balancing and overall system efficiency. Further advancements in renewable energy technologies will be driven by new materials systems that emphasize eco-friendly and productive manufacturing processes. For example, lightweight composite materials for wind turbine blades and advanced photovoltaic devices. In addition, the production of green hydrogen and the use of fuel cells will play an important role in the decarbonization of hard-to-electrify sectors and thus make further important contributions to the creation of a diversified energy system [73,77].

### 3.5. Assessments of Transitions

Technological transitions are complex and multifaceted, requiring a combination of quantitative and qualitative indicators to effectively assess their progress. Many sustainability assessment frameworks and tools currently exist, but significant gaps remain in fully capturing the complexity of ongoing transitions. The following are some indicators that serve as broad metrics to assess the effectiveness and progress of transition efforts across various sectors, encompassing the environmental, economic, and social dimensions of sustainability.

#### 3.5.1. Current Indicators for Assessing Sustainability in Technological Transitions

With the global shift towards more sustainable technological systems, robust indicators that effectively measure the progress of sustainability transitions are essential. The qualitative and quantitative indicators currently used highlight areas that require further attention and improvement. Indicators can be categorized into three distinct groups: environmental, economic, and social [83].

**Environmental indicators**: Technological transitions often drive advancements in reducing emissions, promoting cleaner production processes, and improving waste management. Progress in these areas can be quantified through indicators such as greenhouse gas emissions, overall energy consumption, and the impact on natural resources, including emissions to land, water, and air. Environmental indicators also assess how efficiently land and resources are used and managed. For example, metrics such as use of renewable resources, reduction in raw material extraction, and recycling rates provide valuable insights into the sustainability performance of a given technology [84,85].**Economic indicators**: Economic indicators are essential for assessing the financial feasibility of sustainability transitions. Profitability indicators, such as Return on Investment (ROI), Net Present Value (NPV), and Economic Value Added (EVA), help evaluate the financial returns generated by ongoing transition processes. Productivity indicators focus on elements such as labor and capital productivity, as well as resource efficiency. Additionally, stability indicators, such as financial leverage and economic resilience, aid our understanding of risk management in the context of sustainability transitions [86,87,88,89].**Social indicators**: Social indicators focus on the well-being of individuals and society as a whole. Recently, increasing pressure from stakeholders has forced the integration of social aspects into sustainability assessments. These revised indicators encompass aspects such as acceptance, accessibility, public health, and environmental justice [87,88,90,91,92].

#### 3.5.2. Gaps in Existing Sustainability Assessment Indicators

Although a wide range of indicators already exist for assessing the sustainability of technological transitions, significant gaps and deficiencies remain in the following key areas:**Lack of procedural indicators**: Procedural indicators are useful for understanding the long-term impacts of technologies. They help track significant turning points in technology use, align sustainability transitions with cultural norms, and give direction for developing proactive approaches. By offering continuity and clarity, procedural indicators encourage stakeholder alignment around long-term objectives. Moreover, procedural indicators encourage proactive approaches to difficult transitions by integrating future possibilities, enabling stakeholders to better identify opportunities and mitigate potential risks. Importantly, procedural indicators also foster collaboration and increase accountability, which is vital for achieving inclusive and long-lasting results [93,94].**Overlooked role of windows of opportunity**: Sustainability assessments often overlook the importance of windows of opportunity that can markedly accelerate transition processes. Sustainability assessments typically concentrate on long-term objectives and static indicators, but tend to neglect time-dependent elements such as political upheavals, scientific discoveries, or environmental emergencies. By ignoring these elements, sustainability plans run the risk of stagnating, encountering opposition, and losing opportunities for widespread social alignment. Furthermore, without mechanisms to detect and respond to these opportunities, sustainability plans are less flexible and less reactive to changing problems and new solutions [93,95].**Assessing long-term interactions**: The role of long-term interactions is crucial for ensuring the directionality of transitions. When stakeholders from different domains collaborate towards a shared goal, they develop stronger connectivity and alignment towards common objectives. Although the long-term benefits of transitions are mostly uncertain, with proper assessments and studies, we can capture the persistence and stability of long-term interactions [96,97].**Global supply chains and critical materials**: The transition to renewable energy technologies, particularly in the areas of batteries, solar panels, and wind turbines, relies heavily on critical raw materials such as lithium, cobalt, and rare Earth metals. Current sustainability frameworks often do not assess the supply chain risks associated with the sourcing of these materials, especially concerning their geopolitical implications and environmental impacts [98]. A more comprehensive assessment would consider the environmental and social impacts of material extraction, including the potential for disruptions in global supply chains [99].**Social and distributional impacts**: Many sustainability assessment frameworks focus on environmental and economic outcomes but overlook social dimensions. A number of issues, such as the distribution of costs and benefits across different demographic groups, are not adequately captured by indicators. For example, while metropolitan areas benefit from a steady supply of renewable electricity, rural communities may be subject to the environmental effects of wind farms, without equivalent benefits. Inequality breeds conflict and resistance. Taking these factors into consideration can strengthen sustainability assessments [100].**Isolated use of economic indicators**: Relying on economic indicators to assess sustainability is problematic, as such indicators prioritize financial outcomes over more pressing environmental and social impacts. Economic indicators also tend to focus on short-term gains while overlooking the long-term issues of resource depletion, pollution, and injustice. Furthermore, economic success often undermines environmental damage ignoring negative externalities such as emissions. A broader approach is needed to balance economic, environmental, and social factors for further sustainability transitions [101]. Since sustainability transitions involve numerous emerging technologies, more comprehensive methods are also needed to effectively compare these with mature technologies and provide deeper insight into their economic implications [102].**Lack of useful metrics**: To address the shortcomings of conventional economic indicators in sustainability assessments, it is essential to incorporate useful metrics that account for environmental and social dimensions. Indicators can also include genuine progress indicators, i.e., adjusting Gross Domestic Production (GDP) by considering factors like income inequality, environmental impacts, and the value of unpaid work. Furthermore, integrating the concepts of absolute sustainability assessment into evaluation of technological transitions can provide insights into resource consumption relative to the carrying capacity of the planet [103,104]. To further capture the balance between economic growth, environmental stewardship and social equity, indicators like Social Return on Investment (SROI) can also be utilized [105].

## 4. Analogies Between Biological Evolution and Technological Transitions

Devezas [106] claims that the world of technology is full of biological metaphors that describe an alleged analogy between technical and biological evolution. Additionally, Arthur [26] states that describing technology using biological terms, such as “self-configuring” or “self-healing”, indicates that technology is becoming more biological. Our results support Devezas’ [106] and Arthur’s [26] statements that insights from biological evolution offer valuable frameworks for understanding and assessing technological transitions. In our comparative study, we found that models of both the MLP (Section 4.1) and path dependency (Section 4.2) can be attributed to biological evolution. Additionally, to identify differences, similarities, and commonalities, we compared the characteristics, steps, and flow of the two processes (Section 4.3).

### 4.1. Multi-Level Perspective Attributed to Biological Evolution

Drawing an analogy to the MLP of socio-technical transition presented in Section 3.1, a Multi-Level Perspective can also be adopted for biological evolution. In this context, we can make some general attributions. Landscape of the MLP can be attributed to the ecosystem, encompassing both biotic and abiotic factors. Landscape developments, particularly disruptive events such as asteroid impacts or volcanic eruptions, can be seen as environmental crises that drive change. Triggered by a destabilization of the system, a window of opportunity is created that allows an accelerated transition to a new technology or, in biology, to the formation of new ecological niches or the filling of vacant ones (cf. Section 2.1.4). Niche innovations can be attributed to novel traits of either one taxa (apomorphy), such as mammary glands in mammals, or novel characters shared by two or more taxa (synapomorphy), such as four legs in tetrapods.

Figure 6 shows the evolution of vertebrates based on the representation of the MLP for socio-technical systems (cf. Figure 4). Here we will focus only on the evolution of the dinosaurs, which dominated living nature during the Cretaceous period and became extinct about 66 million years ago [107]. Scientists debate two hypotheses for the mass extinction of non-avian dinosaurs. First, a sudden extinction caused by a huge asteroid impact. Second, a gradual decline in dinosaur biodiversity caused by volcanic eruptions [108]. In any case, the mass extinction of non-avian dinosaurs left a large number of ecological niches vacant, and opened a “window of opportunity”, which caused radiation of other animal taxa filling the vacant ecological niches or forming new ones [109].

### 4.2. Path Dependency Attributed to Biological Evolution

The model of path dependency in technological transitions, as outlined in Section 3.3, can also be applied to biological evolution. The mathematical terms provided by Solée et al. [29] to describe this phenomenon also apply to biological populations. Drawing an analogy to the phases defined in Figure 5, we can identify four phases that describe key principles of biological evolution in the context of speciation (Figure 7). To illustrate these phases, we return to the example of the different flower colors that we used to introduce the principles of biological evolution in Section 2.1.

**Population**: Phase I begins with the gene pool of a population containing individuals with different phenotypic traits, such as blue and red flowers (see Figure 1a). Because of small changes in environmental conditions, some individuals are slightly better adapted than others (shown in the gray shaded area).**Natural selection**: In phase II, the environmental conditions change markedly, triggered by either an abiotic selection factor (e.g., temperature, light, water, humidity) or a biotic selection factor (e.g., predators, competitors, pollinators), affecting organisms in the ecosystem. The selection factor is analogue to the critical juncture shown in Figure 4. As stated in Section 2.1.5, natural selection does not act on a single trait, but always on the whole individual and its totality of phenotypic traits. In our example, we can assume that red flower coloration provides an advantage in attracting pollinators.**Decrease of genetic variability**: In phase III, a decrease in genetic variability occurs, caused by either reproductive isolation (cf. Figure 2) or genetic drift (cf. Figure 3). In analogy to the lock-in shown in Figure 4, these random events can lead to a maximum restriction of the gene pool as described in Section 2.1.3. In our example, only plants with red flowers remain. Devezas [106] states that early decisions in technology can create a path that limits future options, similar to how certain traits in organisms can restrict their evolutionary direction [110].**Increase of genetic variability and restart of further generations**: Phase IV depicts an increase in genetic variability, analogous to the un-locking event shown in Figure 4. As introduced in Section 2.1.2, mutation and recombination are random changes in genetic information and, in our example, result in the additional flower colors—yellow and magenta (cf. Figure 1). Since biological evolution never stops, the plant population exhibiting red, yellow, and magenta flowers will go through further evolutionary cycles.
Figure 7Model of path dependency applied to biological evolution. Colored stars represent existing phenotypes of flower colors with their respective genotypes, while gray stars represent non-existing phenotypes. Phase I is a completely open space representing the gene pool of the population, where the flower colors in the gray shaded area are more likely to occur. In phase II, the red phenotype gains an evolutionary advantage with respect to a biotic or abiotic selection factor (shown as a line). In phase III, due to isolation or genetic drift, the variability decreases to red phenotypes only. In phase IV, genetic changes lead to the appearance of yellow and magenta flowers. The model then restarts from the beginning with this population.
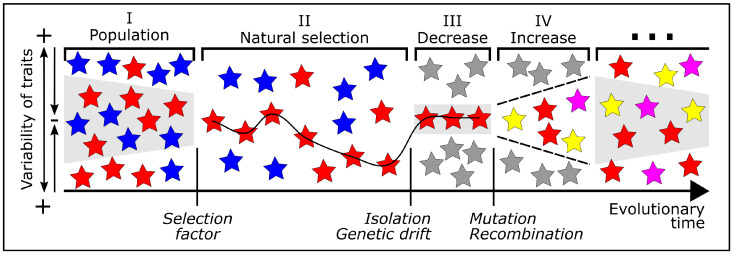


### 4.3. Differences, Similarities, and Commonalities

Building on the identified analogies, we will now compare biological evolution with the technological transitions to identify differences, similarities, and commonalities between the two domains. Initially, the comparison has led us to the following fundamental observations:**Theory versus mission statement:** In contrast to a theory, which is defined as a system of scientifically based statements suitable for explaining regularities and making predictions about the future, a mission statement is a written declaration that describes strategies and actions to achieve a formulated target state. The classical theory of evolution refers to the changes in the genetic material of all organisms over time [16,30]. This theory is continuously expanding through ongoing scientific findings and is therefore now referred to as the “modern synthesis” [31]. In contrast, sustainable development is guided by the teleological and anthropocentric mission statements, such as the SDGs formulated in the 2030 Agenda [6], and the policy-driven goals of technological transitions set by governments (cf. Section 3.4.1).**Evolutionary time versus time:** While biological evolution and technological transition are both processual, they differ tremendously in terms of time scale. Biological evolution takes place over evolutionary timescales spanning millions of years, while technological development occurs over much shorter periods, ranging from months to years.

Based on the comparison, we derived key lessons learned from biology to better understand and manage transitions in the technosphere, as well as suggestions for qualitative and quantitative metrics and indicators:**“Trial and error” versus targeted development and market acceptance:** A key difference between biological evolution and technological transition lies in change and selection. In biology, this corresponds to the random change in genetic variability together with natural selection. In contrast, technological change is typically intentional and purpose-driven by engineers [27], with selection occurring through market dynamics [111]. For example, just as natural selection favors organisms best suited to their environments, technologies that meet market demands and societal needs are more likely to succeed [106]. The “trial and error” approach has been highly effective in the context of Evolution Strategies: a powerful tool for solving minimization and maximization problems by generating a larger set of optimized solutions. The lesson learned for technological transitions is “Give it a try”. A suitable indicator in this respect should cover the proportion of successful niche innovations relative to the total number of niche innovations.**Continuous versus disruptive process:** The evolutionary perspective emphasizes both steady progress and sudden, disruptive changes [26,106], two types of progress that we also find in technological developments. Despite the difference in time scales, generation-based biological evolution and version-based technological development are comparable. In biology, continuous progress is maintained because characteristics must be inherited from one generation to the next and generations cannot be skipped. However, detrimental mutations or disruptive environmental conditions can lead to the extinction of entire taxa. In contrast, technological development allows for versions to be skipped, for example due to teething problems or disruptive events, such as the transition from analog to digital devices. Thus, lessons learned from biology are “Do not close for reconstruction” and “Keep older versions in the innovation process”. Indicators for the sustainability assessment of transitions should therefore include the permanent supply of key functionalities by the regime, as well as the diversity and accessibility of older technological versions of a technology.**Increase of variety:** In biology, an increase in genetic variability refers to random changes in genetic information through mutation and recombination, which can result in new phenotypes of a trait. In technology, the concepts of mutation and recombination translate into the encouragement to create niche innovations by thinking beyond existing constraints or limitations of market or consumer segments. A suitable qualitative indicator in this context should reflect the degree of novelty of the niche innovation, particularly in cases where no established standards exist and the new functional unit cannot be fulfilled by any existing application. However, a limitation of this approach is the risk of creating innovations that offer limited value or relevance to consumers, and this trade-off should be kept in mind.**Decrease of variety:** In biology, a decrease in genetic variability refers to the random changes in genetic information due to isolation and genetic drift. This occurs when phenotypes of a trait disappear completely or when initially rare phenotypes become much more frequent. In technology, a reduction in variety can result from exnovation, which is an effective approach to overcoming path dependencies. Exnovation can be described as a deliberate process of phasing out or terminating outdated technologies. It can also include products, practices, or systems within the technosphere or society that are incompatible with the goals of the transition process. As the opposite of innovation, exnovation involves the removal of unsustainable elements within the regime in order to open the field for more efficient, consistent, or sufficient solutions, e.g., in the form of niche innovations [93]. Successful exnovations can be measured by the adoption rate of new technologies, a metric that determines how quickly new, more sustainable technologies are implemented as a result of an innovation.**Phenotypic and technological convergence:** In biology, convergent evolution describes the development of similar phenotypic traits in unrelated taxa, arising from different initial structures under similar environmental selection pressures. In technology, convergence refers to the process of integrating previously separate or unrelated technologies into unified, more advanced products, services, or systems that offer combined functionalities in a more efficient platform. The similarity between phenotypic convergence in evolutionary processes and technological convergence led us to hypothesize that pressures during the innovation process, e.g., user needs, market demands, and regulatory environments, can act as catalysts, driving technological convergence and promoting adaptive optimization of emerging technologies. A particularly striking current example is the rapid convergence of previously unrelated technologies such as the internet, cameras, and navigation systems, into multifunctional smartphones. Technological convergence can be assessed through indicators measuring the co-occurrence of technologies in scientific publications and patents, revealing how often a technology is referenced across different scientific disciplines. Additionally, a strong conversion potential may be indicated by the interdisciplinarity of patents, measured by the range of different technology areas citing a given patent.**Fitness and consistency:** According to Darwin’s original framing, fitness refers to individuals being better adapted to their immediate and local environment than other organisms. Fitness can be measured by the reproductive contribution of an organism relative to the contribution of other organisms in its population (see Section 2.1.5). Similarities to this concept can be found in technologies under path dependencies, when the defining components of a regime continually and dominantly reproduce themselves, limiting opportunities for upcoming niche innovations. Fitness can also be compared to the major sustainability strategy of consistency. In this context, consistency refers to the integration of a technology and its associated material and energy flows into natural cycles, reflecting the principles of circularity and waste minimization [9]. From this perspective, we conclude that any innovation that contributes to an ongoing transition process must be compatible with the natural limitation of the ecosphere and should ideally integrate into natural cycles. Ultimately, consistency requires innovation to operate within the natural limits as defined by the Planetary Boundaries. Therefore, a suitable indicator for evaluating such innovations should measure the degree of alignment with the Planetary Boundaries.

### 4.4. Outline of a Concrete Indicator for Assessing and Monitoring Sustainability Transition Processes

Inspired by the comparison between the determinants of biological evolution and technological transition, we have derived a set of ideas for metrics and indicators to assess and monitor sustainability transition processes. These ideas serve as a starting point for defining and specifying metrics and indicators. However, the proposed metrics and indicators outlined require further development and refinement to become fully operational. A selected example will be used to illustrate how this process of development can be carried out.

One particularly interesting similarity was found between “fitness” and “consistency”. As pointed out in the previous section, a suitable indicator must measure the degree of alignment with the Planetary Boundaries. To assess whether an innovation is consistent with the Planetary Boundaries, it is essential to assess its impact on the nine critical Earth system processes (e.g., climate change, loss of biodiversity, biogeochemical cycles), as defined by Steffen et al. [103]. The assessment can be conducted using a Life Cycle Assessment (LCA) approach [112] that evaluates the life cycle impacts of an innovation against safe operating limits [113,114]. The result is a set of quantitative metrics, each corresponding to one of the nine Earth system processes, specifying whether and to what extent the Planetary Boundaries are affected by the innovation, e.g., through inventories of greenhouse gas emissions that exceed the climate change boundary.

To provide a broader picture of sustainability performance, the environmental metrics outlined above should be joined by economic aspects. Relevant aspects include the innovation’s ability to improve target sector resilience, i.e., their ability to withstand economic disruption, including the diversification of supply chains and adaptation to new regulations. Furthermore, in order to overcome gaps and deficiencies of existing indicators sets related to sustainability assessment mentioned in Section 3.4.2, procedural, social, and distributional aspects should also be incorporated. Suitable metrics in this respect might include the degree of public acceptance towards the innovation, as well as the involvement of stakeholders during its design process.

Once the individual metrics have been determined, we propose to aggregate them into an overarching indicator, which could be termed the “Absolute Sustainability Compliance Rate (ASCR)”. To calculate the ASCR, all metrics are normalized to a uniform numerical scale (e.g., from 0 to 1, where 1 represents the highest level of compliance) and subsequently aggregated into a single value. An ASCR of 1 indicates that the innovation fully meets the principles of absolute sustainability [115], because it operates within the Planetary Boundaries while also achieving economic and social targets. A score below 1 would suggest room for improvement.

Due to its action-oriented approach, the ASCR indicator is intended to be implemented at the level of product development and as part of research and development cycles. Specifically, it is designed to provide guidance for innovation processes within businesses and research organizations that aim to generate sustainable technological solutions and can be integrated into strategic planning tools, such as TAPAS (see Section 1.2).

## 5. Discussion

Biomimetics is associated with the hope of learning lessons from living nature and applying them to technical challenges in order to find adequate solutions in technology using approaches that have evolved and have been proven in biology over an evolutionary time scale. In recent years, biomimetic products have been successfully developed in various fields of application by transferring the functional principle and its underlying structures of the selected biological model to a technical product [116]. A prime example among these successful biomimetic developments is the self-cleaning leaf surface of the sacred lotus (*Nelumbo nucifera*) [117], whose functional principles have been transferred to technical surfaces marketed under the brand name Lotus-Effect^®^. One such success story is the development of the self-cleaning facade paint Lotusan^®^ [118,119]. A comparative sustainability assessment of the biomimetic facade paint Lotusan^®^ and the conventional paint Jumbosil^®^ showed the biomimetic paint to be a cost-effective and resource-efficient alternative [120].

Much more challenging, however, is the attempt to identify biological models developed during biological evolution for teleological and anthropocentric mission statements, such as sustainable development or technological transition. Since these mission statements are processes, biological evolution seems at first glance to be a very suitable model. This perspective is also reflected in some current literature on transition research [26,106,111], although sometimes without providing scientific proof [121], an omission that this publication seeks to address.

In fact, such a transfer is highly challenging. Although both biological evolution and mission-driven transitions are process-based, they have fundamental differences. Biological evolution differs markedly from the mission statements because it is neither teleological nor anthropocentric. The key principles of biological evolution, namely the random increase or decrease in genetic variability at the individual and population level and natural selection, are incompatible with the intentional and goal-oriented working methods of engineers, politicians, governments, and public decision makers. The contrast becomes even more apparent when AI-based software with stochastic algorithms (e.g., Evolution Strategy) is used to develop new published by products. Devezas [106] advocates Digital Darwinism as a new science based on further, still lacking, enhancements of genetic algorithms and genetic programming, such as the transmission of information as a common denominator across all growth and diffusion phenomena, to create the knowledge base for the process of technological change.

The economist and complexity theorist W. Brian Arthur [26] compares Darwin’s question of how novel species arise with the equivalent question of how radically novel technologies emerge. He coined the term of “combinatorial evolution”, which implies that inventions result from the intentional combination of existing technologies through a process involving the interplay of experience and knowledge, driven by necessity [28]. According to Arthur [26], radically novel technologies do not emerge through the accumulation of small changes to previous technological products favored by natural selection. Instead, technology is “self-creating”, meaning novel technologies emerge from combinations of existing ones and, in turn, become potential components of future technologies [27]. Due to unpredictable factors—such as which phenomena will be discovered and transformed into the basis for new technologies, which combinations will emerge from a vast number of possibilities, or when a window of opportunity will occur—the development of the product is historically contingent. This does not mean that technological development is completely random. However, the future of technology is not entirely predictable, just as the future of biological species is unpredictable based on the current collection of species. Nevertheless, biological evolution and technical development share one thing in common: both follow an ancestral lineage, which we refer to as generation-based and version-based development. Solée et al. [29] have adopted the view that technological innovations are highly dependent on the combination of pre-existing elements, a notion nicely illustrated by the evolutionary tree of cornets. In line with our view, populations play a major role in the diversification, convergence, and extinction of properties in both systems. However, planned design in technology has no equivalent in biological evolution.

Against this background, Speck et al. [9] were able to sharpen the sustainability strategies of efficiency, consistency, and sufficiency by attributing biological concepts found in the lianescent growth form. For example, the biological concepts of “lightweight construction” and “modularity” can be attributed to efficiency, the concepts of “zero waste” and “best fit” to consistency, and the concepts of “less is more” and “good enough” to sufficiency.

A particular challenge lies in applying “lessons learned from biological evolution” to processes in the technological transition, which reflects our first scientific question (Section 1.4). From a biologist’s perspective, some attributions may appear intuitive, such as attributing mutation and recombination to innovation, or gene drift to exnovation. However, the inherently random nature of evolutionary processes remains an obstacle, as it contrasts the deliberate, goal-oriented decisions claimed to be made by engineers and policymakers. This is all the more surprising because the optimization method “Evolution Strategy”, which deals with minimization and maximization problems, accepts random changes in the parameters and produces innovative solutions (cf. Section 2.2). However, if the parameters are changed randomly, we must accept that the results will fluctuate between better and worse in the course of the optimization process. These fluctuations are not a major problem when the Evolution Strategy is applied in the context of development processes, as the Evolution Strategy is a robust method with algorithms that eventually lead to an optimal solution, even if unforeseen faults, errors, or mistakes occur. In the context of the energy transition, a meaningful real-world example is the study published by Dujardin et al. [50] showing that renewable electricity generation can be effectively placed and shared with existing electricity generation while respecting the constraints of the electrical grid. In recent years, some scientists have favored evolution management as another real-world example. In this case, management strategies for the organization of companies [122] and economic aspects of projects [123] are derived from the principles of biological evolution.

From an engineering perspective, interesting insights can be gained by comparing biological evolution and technological transitions. Niche innovations are particularly interesting in this context. In contrast to the random increases and decreases in genetic variability, innovations in the technosphere usually do not arise by chance; rather, they are at least partly deliberately induced by their creators within the limits of their capabilities [121]. Nonetheless, it can be beneficial to just “give it a try” (cf. Section 4.3), while allowing for random changes in key parameters during development. In addition, biological evolution teaches us not to discard information that is not currently needed (e.g., redundant alleles), but to pass it on to the next generation of offspring. In the context of technology, this means “keeping older versions of a technology in reserve”. Even if older versions appear ineffective or inefficient at present, they may become part of the solution again under different conditions. Overall, storing older versions can increase the diversity of possible paths to sustainability, thereby reducing the risk of new path dependencies or lock-in effects that can arise in the transition process. Ultimately, however, the viability of any possible solution must be tested within the mainstream, with an existing or newly emerging regime serving as the “selection environment” in which niche innovations must prove their viability.

With regard to our second scientific question (cf. Section 1.4), we were able to derive several metrics and indicators inspired by biological evolution. The metrics and indicators can serve to enhance our understanding of ongoing transition processes, and help monitor the sustainability of materials systems and technologies emerging in their context (cf. Section 4.3). As demonstrated in Section 4.4, an integrated indicator for absolute sustainability assessment can provide a promising approach for assessing the sustainability of innovations and emerging technologies on a systemic level, capturing environmental, economic, and social aspects.

More broadly, it is important to emphasize that technologies emerging from technological niches are not ends in themselves. To succeed, they must also contribute to regime change. When the goal is to move innovations beyond the niche, it is necessary to coordinate existing niche activities with developments at the regime level, or to establish co-evolutionary processes between the two. The example of the energy system transition makes this point clear: it is not enough to develop niche innovations alone—these innovations must also be translated into regime alternatives. Within this context, we must keep in mind that transformations on a societal level are not solely driven by technical innovations. When adopting a multidimensional regime perspective and taking an evolutionary understanding of innovation seriously, we realize that non-technological elements of the regime, such as novel governance approaches and changes in consumer habits, also emerge through innovation processes and need to be cultivated within niches as well [121]. A model example of legislative support for a technological niche innovation is the Solar Package (“Solarpaket I”) introduced by the German Federal Government in 2024. This initiative aimed to make balcony power plants more attractive to private households. In addition to a simplified registration process and improved plug-and-play features, the Solar Package included raising the power limit to further improve the cost-effectiveness of the plants [124]. As a result, installing balcony power plants in Germany has become less bureaucratic, technically easier, and more affordable. What started as a niche innovation and hobby for technology enthusiasts became a mainstream solution for many households, marking a significant milestone in Germany’s energy transition.

## 6. Conclusions and Outlook

Living nature in general, and biological evolution in particular, appear to be appropriate models for addressing technical challenges, such as the process of technological transition. The scientific theory of biological evolution is grounded in two key principles: random change of genetic variability and natural selection. Based on these key principles, the biomimetic optimization method “Evolution Strategy” was developed, aiming to find innovative solutions for technical challenges. Furthermore, we are convinced that modeling biological and technological evolution in an interdisciplinary project can provide further insights into analogies and disanalogies.

Although applying processes from biological evolution to teleological and anthropocentric missions, such as sustainable development or technological transitions, remains a complex challenge, we identified several approaches to better understand and inspire the management of ongoing transition processes. Particularly worth mentioning is the approach of increasing the diversity of niche innovations, both by deliberately introducing random changes in key development parameters and by keeping older versions of a technology in reserve. Together with corresponding metrics and indicators, we have laid the foundation for action-oriented guidance on developing more sustainable technological solutions in major ongoing transitions, such as the energy transition.

In terms of further research, the metrics and indicators outlined in this paper must be further developed, concretized, and operationalized. Once accomplished, they will be used as additional indicators within the methodological framework of TAPAS [7], as introduced in Section 1.2. TAPAS is intended as a strategic planning tool within the innovation process and is therefore ideally suited for implementing these indicators into the decision-making processes of ongoing research and development projects conducted by businesses or research institutions. The new indicators will enable an expansion of TAPAS to include the assessment of material systems and their potential applications, particularly with regard to their ability to support sustainability transition processes in key industrial sectors.

## 7. Glossary

We have compiled a list of key terms from the fields of biology, sustainability research, and technological transitions to increase the readability of the article.

**Adaptive radiation:** In biology, adaptive radiation is the rapid diversification of a single lineage into organisms forming different niches [39].**Allele:** In biology, allele are Variants of a gene (=variants of the sequence of nucleotides on a DNA molecule) at a specific gene locus of a chromosome [17].**Anthropocentric:** From a philosophical perspective, anthropocentric considers human beings as the most significant entity in the universe, prioritizing human interests and values above those of other species and the environment [125].**Biomimetics:** Biomimetics is the transfer of a functional principle derived from living organisms into a technical application [116,126].**Biological evolution:** Biological evolution is the genetic change in a population from generation to generation [17].**Bottleneck effect:** In biology, the bottleneck effect is the drastic reduction of a once large population to a few individuals, usually caused by a natural disaster [17].**Co-evolution:** In biology, co-evolution describes the evolution of traits of interacting unrelated taxa [127]. In technological transition, co-evolution refers to shared work of generating innovative and exceptional design conducted by various actors from firms, customers, and collaborating partners [93].**Consistency:** Consistency is a major sustainability strategy that pertains to the integration of a technology and related material and energy flows into natural cycles, reflecting the principles of circularity and waste minimization [9].**Demand Response System:** In technology, demand response refers to balancing the demand on power grids by encouraging customers to shift electricity demand to times when electricity is more plentiful or other demand is lower, typically through prices or monetary incentives [128].**Diversity:** Biodiversity means the variability of life existing on Earth, which can be measured as genetic variability, phenotypic diversity, species diversity, and ecosystem diversity [129]. In technology, diversity refers to the state of having or being composed of differing elements [130].**Diversification:** In evolutionary biology, diversification refers to the development of differences in speciation [131]. In technology, diversification refers to the expansion of a technologies competence [132].**Ecosphere:** A comprehensive system that encompasses all living organisms and their environmental interactions on Earth [133].**Ecosystem:** In biology, an ecosystem includes all the organisms in a given area and the physical and chemical environment in which they live [17].**Efficiency:** A measure of the ability to achieve maximum output with the least amount of input, particularity in terms of resource use such as energy and materials [9].**Environment:** In biology, environment is the sum of extrinsic potential selective factors encountered by a population or species, including habitat, climate, and sympatric species, among many others [134].**Evolution Strategy:** Optimization method based on the key principles of biological evolution [25].**Exnovation:** Process in which a given technology is currently no longer used because its physical infrastructure has been deliberately removed [135].**Fitness:** In biology, fitness describes the capacity of an organism to survive and transmit its genotype to reproductive offspring as compared to competing organisms. Fitness can be quantified by the contribution of an allele or genotype to the gene pool of subsequent generations as compared to that of other alleles or genotypes [136].**Founder effect:** In biology, the founder effect describes the random changes in allele frequency as a result of the founding of a new population by very few individuals [17].**Functional unit:** In sustainability assessments, the functional unit is defined as the quantity of a product or product system on the basis of the performance it delivers in its end-use application [112].**Genetic drift:** In biology, genetic drift describes the changes in allele frequencies in a small population from one generation to the next as a result of random events [17].**Generation-based development:** In biology, genetic information is passed from the parental generation on to the next offspring generation [17]. In Evolution Strategy, information about variables is passed from the parental generation on to the next offspring generation [25].**Genetic variability:** In biology, genetic variability represents the genetic differences within or between populations [137].**Genotype:** In biology, the genotype is a precise description of the genetic material of an individual, either for a specific trait or for a range of traits [17].**Gene pool:** In biology, a gene pool is the total number of alleles of all genes from all individuals in a population at a given time [17].**Goal:** A goal is a general statements of what is to be achieved, but without giving precise figures or setting deadlines for completion [9].**Indicator:** An indicator is a qualitative or quantitative aspect or metric that expresses achievement or progress towards a specific goal or strategy [138,139].**Landscapes:** In biology, a landscape is a part of the Earth’s surface that forms a spatial unit through its inorganic (e.g., surface form, rock, soil), organic (e.g., animals, plants, fungi) and anthropogenic (e.g., agriculture, industry, housing) components [140]. In a maximization or minimization problem, a three-dimensional quality landscape consisting of mountains of different heights and valleys of different depths is created by plotting two parameters horizontally and the corresponding value of the variable to be optimized, the quality, vertically [25]. In MLP, a landscape is an external structure or a setting for interactions of various actors in a technological system [52].**Lock-in effect:** In the context of path dependency, the lock-in effect is a stage in systems where certain technologies, practices or policies have come to stay making it difficult to shift to more sustainable alternatives [62].**Material and energy flows:** In industrial systems, material and energy flows refer to material- or energy-related inputs or outputs from a process or product system [112].**Metric:** In the context of sustainability, metrics are the building blocks for measuring the environmental, economic, and social aspects of a system, while certain aspects, like greenhouse gas emissions, can be measured uniformly, many sustainability issues (e.g., related to social aspects) are complex phenomena that require proxies to simplify and quantify them [138,141].**Mutation:** In biology, mutations are stable and heritable random changes in genetic material. One allele can mutate and become another allele [17].**Natural selection:** Darwin understood natural selection as differential reproductive and survival success of offspring that fit best to the natural environment [17].**Niche:** In biology, the ecological niche describes the set of relationships between a species and its environment [142]. In technology, a niche is a specialized market segment or application area where a particular technology is uniquely suited to meet specific needs or solve distinct problems. Niches often serve as early adoption environments for emerging technologies, allowing them to improve at a small scale, and, under the right conditions, challenging and eventually replacing the dominant regime [59].**Paradigm shift:** A process that intensifies over time and causes fundamental and irreversible changes in the regime [61].**Path dependency:** A phenomenon that describes how historical decisions and established practices shape the trajectory of future developments, often leading to a reliance on specific technologies or systems that can hinder innovation and ongoing transitions [62].**Phenotype:** In biology, a phenotype is an observable characteristic of an individual resulting from the influence of both genetic and environmental factors [17].**Phenotypic convergence:** In biology, convergent evolution is the independent evolution of similar phenotypic traits in unrelated taxa from different initial structures under similar selection pressures [17].**Planetary Boundaries:** The nine biophysical systems and processes that govern the operation of Earth’s life support systems and, ultimately, the stability and resilience of the Earth system [103].**Population:** In biology, a population is a group of individuals of the same species living in the same habitat at the same time and capable of reproduction [17].**Prosumer:** An individual who both consumes and produces [143].**Recombination:** In biology, recombination is a random rearrangement of parental genetic material during sexual reproduction [17].**Regime:** Established practices, rules, and institutions supporting dominant technologies and systems. The established alignment of technologies, policies, user patterns, and cultural discourses is also called a socio-technical regime that stabilizes systems like energy and mobility shaped by shared rules and institutions influencing the actions of various social groups, while generally stable, regimes can be disrupted by external pressures or niche innovations [52,144].**Reproductive isolation:** In biology, reproductive isolation is the interruption of gene flow between populations of the same species, preventing the production of offspring under natural conditions [145].**Resilience:** The ability of a system to withstand, adapt to, and recover from changes or disruptions [146]. In biology, resilience is the ability of a system to manage damage by returning to its original functionality after (partial) failures have occurred [47].**Robustness:** The capability of a system to prevent damage and to maintain its functionality even if unforeseen faults (e.g., in hardware) or errors (e.g., in software) occur or a mistake is made by a human being [147].**Smart grid:** A modernized electrical grid that uses digital technology, communication networks, and advanced sensors to enhance the efficiency, reliability, and sustainability of electricity distribution. It enables two-way communication between utilities and consumers, allowing for real-time monitoring and management of energy resources [81].**Socio-technical system:** A Socio-technical system pertains to the interdisciplinary scientific perspective that examines technological progress by taking into account the relationships between different components, such as technology, laws, customs. In addition to the creation and application of technology, these systems also encompass the processes of production, dissemination, and social integration [144,148].**Sufficiency:** A major sustainability strategy that emphasizes on change in consumption pattern and respects ecological limits by reducing the surplus use of resources [9].**Sustainable Development Goals:** A set of 17 global objectives established under the United Nations’ 2030 Agenda for Sustainable Development. SDGs call for urgent, collective action through global partnerships to address critical challenges, including, climate change, environmental degradation, poverty, inequality and access to justice [6].**TAPAS: T**iered **A**pproach for **P**rospective **As**sessment of Benefits and Challenges is a sustainability assessment methodology designed to be used during the development process of novel technologies and materials systems. TAPAS uses a systematic approach to guide inquiry, provide instructions and recommendations to evaluate and enhance the sustainability performance of the objects examined [7].**Target:** A specific and measurable result to be achieved within a given time period [9].**Technological convergence:** The process of integrating previously separate or unrelated technologies to create new, more advanced products, services, or systems that combine the functionalities of multiple technologies into a single, more efficient platform [149].**Technological transition:** Systemic changes that reshape socio-technical systems like energy, transportation, or communication. The changes arise from innovations that disrupt established systems and shift social, economic, and regulatory structures [52].**Technosphere:** A system composed of human-made technological and infrastructural elements that interact with the Earth’s environment, biosphere, and society. It includes all human-made systems, devices, procedures, and networks developed and operated by humans to support contemporary society [150].**Teething problem:** Temporary problem connected with a new product or at the beginning of a process/activity that occurs when a new technology is being fine-tuned and adapted for optimal performance [151].**Teleological:** Teleological means starting from the end and reasoning back, explaining things based on their goal or end purpose [152].**Version-based development:** A systematic approach to creating and managing sequential iterations of a product, software, or technology. Each version represents a distinct stage in the product’s development process, incorporating new features, improvements, bug fixes, or performance enhancements [153].

## Figures and Tables

**Figure 4 biomimetics-10-00406-f004:**
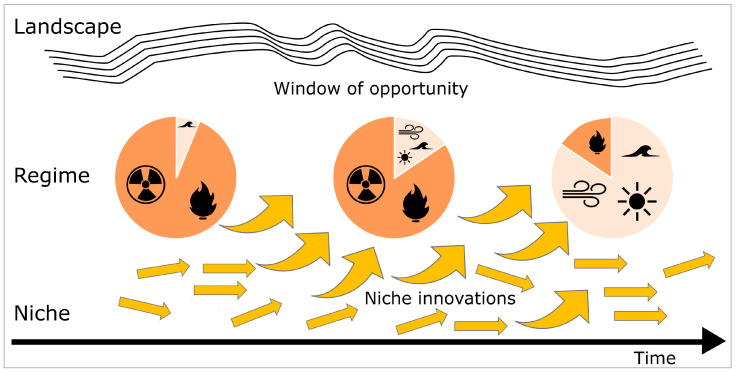
The Multi-Level Perspective with its embedded systems of landscape, regime, and niche is illustrated using the energy sector as an example. The pie charts show the reconfiguration of the energy regime from one dominated by fossil fuels and nuclear power (dark orange) to one dominated by renewable energy sources such as solar, wind, and hydro power (light orange). Adapted from [57].

**Figure 5 biomimetics-10-00406-f005:**
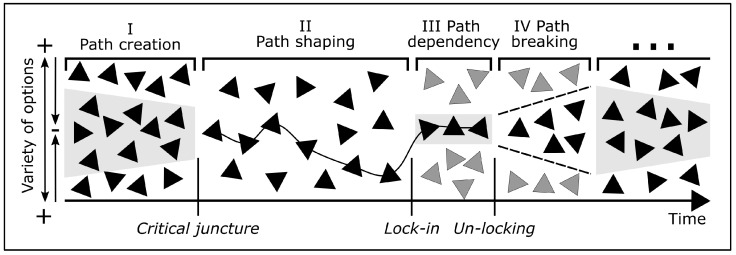
Model of path dependency in technological transition. Black triangles represent available options, while gray triangles represent non-available options. Phase I is a completely open space of possibilities, where the triangles in the gray shaded area are more likely than others. In phase II, a path (depicted as a line) is created as a result of a special event (critical juncture). In phase III, lock-in occurs, which is the special case of maximum restriction of available options. In phase IV, alternative options are available again (un-locking) and the model starts from the beginning. Adapted from [64].

**Figure 6 biomimetics-10-00406-f006:**
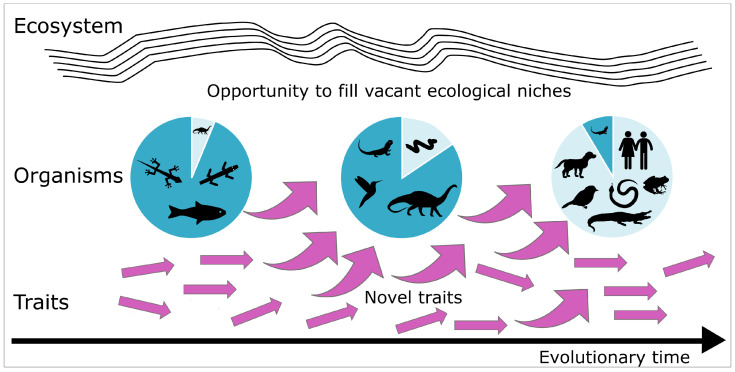
The Multi-Level Perspective attributed to ecosystems, organisms, and traits is illustrated using the evolution of vertebrates as an example. Initially, the world was dominated by early representatives of amphibians, reptiles, and fish (dark blue). Then dinosaurs and birds dominate, while early mammals appear (light blue). After the extinction of the non-avian dinosaurs, more recent amphibians, reptiles, birds, and mammals filled the vacant ecological niches. However, there are still surviving members of taxa that were very diverse in earlier eras, such as the Tuatara, which are often denominated “living fossils”.

**Table 1 biomimetics-10-00406-t001:** Comparison of adaptation in biological evolution and the optimization method Evolution Strategy used in technology (adapted from [47]).

	Biological Evolution	Evolution Strategy
Subject	Living being	Object to be optimized
Mutation	Random change of genetic information	Random change of input variables (i.e., object parameters)
Recombination	Reshuffling of parental genetic material (e.g., meiosis)	New combination of parental object parameters
Selection	Selection of those individuals with the best fit to the natural environment	Selection of those individuals that best meet the optimization criterion
Result	Adapted organism	Optimized object

## Data Availability

All relevant data are included within the article.

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
