# Peer review of "Transition Processes in Technological Systems: Inspiration from Processes in Biological Evolution"

_biomimetics, 2025, doi:10.3390/biomimetics10060406_

Round 1
Reviewer 1 Report
Comments and Suggestions for Authors
The authors identified analogies between biological evolution and technological transitions in terms of the Multi-Level Perspective and the path dependency model. They also gained insights into better managing technological transitions and proposed specific indicators for assessing and monitoring these transitions. Given the strong innovativeness of the work, we would like to raise the following points for further improvement:
1.In the Introduction section, it is necessary to summarize the core innovations and highlight them prominently, as this would be valuable in demonstrating the originality of the paper.
2.Is it possible to further validate the relationship between biological evolution and technological transitions through physical or mathematical modeling?
3.In the Discussion section, it would be beneficial to compare the findings with previous studies and summarize how your research outcomes could be applied in practice.
Author Response
General Comments
Comment-R1-01: The authors identified analogies between biological evolution and technological transitions in terms of the Multi-Level Perspective and the path dependency model. They also gained insights into better managing technological transitions and proposed specific indicators for assessing and monitoring these transitions. Given the strong innovativeness of the work, we would like to raise the following points for further improvement.
Answer-R1-01: We thank Reviewer 1 for taking the time to read and assess our manuscript. We have carefully gone through her/his comments and requests and found them all very valuable for further improving our manuscript. Our detailed answers are given in the Author's Notes. Any revision made to the manuscript are marked up in red. Please note that we have removed Figure 1 from the manuscript to shorten Section 2.1. In addition, we would like to point out that the Figures 4 and 6 (Please note: new numbering) have now been supplemented with public domain silhouettes for copyright reasons and thus differ slightly from the first versions. We have adapted the figure captions accordingly.
Detailed comments
Comment-R1-02: In the Introduction section, it is necessary to summarize the core innovations and highlight them prominently, as this would be valuable in demonstrating the originality of the paper.
Answer-R1-02: Thanks for the interesting point about summarizing the main contributions at the end of the introduction. Since we find the idea very intriguing for a conceptual paper, we present our main contributions at the end of the introduction (see page 4, 4th paragraph, lines 168-178).
Comment-R1-03: Is it possible to further validate the relationship between biological evolution and technological transitions through physical or mathematical modeling?
Answer-R1-03: We thank the reviewer for this very interesting question, which prompted us to search for relevant literature and discuss this point intensively. We found literature with a variety of ways to mathematically model biological evolution, but also to find similarities between biological and social evolutionary mechanisms (Arthur 2009, Devezas 2005). If a min-max problem is to be solved, it is possible to apply the evolutionary strategy. In addition, we found some interesting publications that we have included and discussed at the appropriate places in the manuscript (see page 3, 5th paragraph, lines 134-140; page 4, 1st paragraph, lines 141-145; page 14, 1st paragraph, lines 448f.; page 19, 3rd paragraph, lines 713-716; page 20, 1st paragraph, lines 745f.; page 24, 6th paragraph, lines 7956-958; page 25, 2nd paragraph, lines 975-996; page 25, 4th paragraph, lines 1016-1019; page 26, 1st paragraph, lines 1020-1023). For example: the opposite trend of random mutation and rational planning in networks published by Yan et al. 2020 and mathematical terms that describe lock-in and path dependency published by Solée et al. 2013. Unfortunately, it is not possible to go into more detail in the manuscript because it would make the manuscript even longer and reviewer 2 advised us to shorten the manuscript. The literature search on modeling has convinced us that modeling is very suitable for future interdisciplinary projects of biologists, engineers and simulation scientists. For this reason, we have included this aspect in the “Outlook” section (see page 27, 1st paragraph, lines 1076f.).
Comment-R1-04: In the Discussion section, it would be beneficial to compare the findings with previous studies and summarize how your research outcomes could be applied in practice.
Answer-R1-04: Thank you very much for this important note on the contents of the Discussion section. We were happy to take up this point and have compared the results of our research with the findings of previous studies at the interface of biological evolution and technological transition that we have also amended in the Introduction section.
In the Section “Introduction”, we have added information about the concept of “combinatorial evolution”, which was developed by the economist and complexity theorist W. Brian Arthur (2019a, b). Additionally, we have described the study of Solée et al. 2019 regarding commonalities and distinctions between biological evolution and technology development (see page 3, 5th paragraph, lines 134-140; page 4, 1st paragraph, lines 141-145). In the Section “Discussion”, we have again referred to these publications and compared their statements with our results (see page 25, 2nd paragraph, lines 975-996).
With regard to the practical application of the results, we have further clarified how the new indicators can be integrated into TAPAS (see page 27, 3rd paragraph, lines 1091-1094). TAPAS is intended as a strategic planning tool in the innovation process and is therefore ideally suited for implementing the indicators in the decision-making processes of ongoing research and development projects carried out by businesses or research institutions.
Reviewer 2 Report
Comments and Suggestions for Authors
Review:
Transition Processes in Technological Systems: Inspiration from Processes in Biological Evolution
This manuscript explores analogies between biological evolution and technological system transitions, aiming to derive insights for sustainability assessment. It is an ambitious and well-written manuscript, conceptually rich, and the result of substantial work.
I appreciate the authors' effort to initiate such a novel cross-disciplinary dialogue. However, I believe the manuscript, while promising, currently falls short of the depth and coherence needed for publication in its present form. My suggestions below are offered constructively to support further development of what could become a valuable contribution.
Strengths
Ambitious and interdisciplinary—addresses a compelling topic with relevance for biomimetics, transition studies, and sustainability metrics.
Strong conceptual and visual presentation (e.g., use of MLP, path dependency, glossary).
The analogy between evolutionary principles and transition mechanisms is intriguing and thought-provoking.
Areas for Improvement
- Conceptual Focus and Scope -- The manuscript spans evolutionary biology, sustainability theory, system transitions, and implicitly philosophical considerations. While each theme is well-developed, the paper as a whole becomes diffuse and difficult to follow.
Suggestion: Sharpen the conceptual focus. Consider narrowing the scope based on key points you want to contribute to and clarify the intended audience.
- Philosophical Assumptions about Evolution -- The paper treats evolution as purely random, however, this framing oversimplifies ongoing debates in evolutionary theory, especially concerning macroevolution, purpose, and emergence. Since sustainability is inherently teleological, this ontological tension needs addressing.
Suggestion: Acknowledge the limitations of the naturalistic framework and briefly reflect on alternative views (e.g., systems thinking, ontological pluralism).
- Lack of Empirical Grounding -- Despite frequent mention of metrics and indicators, the paper remains purely conceptual. There is no application of the proposed ideas to a case study, nor any operational examples.
Suggestion: Could any minor worked example be included? Or discussion on how the presented Evolution STrategy. This would aid in demonstrating how the insights translate into actionable assessment tools.
- Missing Innovation Dynamics within Firms -- While the paper draws system-level analogies between evolution and technological system change, it misses the rich evolutionary dynamics occurring within product development and R&D cycles—e.g., variation, selection, and abandonment of ideas.
Suggestion: Strengthen the analogy by incorporating more levels of innovative processes than on the macro transition level.
- Readability and Length -- The manuscript is long and concept-heavy. While the glossary and structure help, the core contribution gets somewhat diluted.
Suggestion: Consider reducing length by summarizing background sections and focusing on the novel contributions and implications.
Final Recommendation
I recommend a major revision. The manuscript offers valuable ideas, but it needs clearer framing and stronger philosophical grounding. I would like to thank the authors for an interesting read, which I believe will be improved through valuing the above-mentioned aspects and how these can improve the manuscript.
Some specific comments:
The introduction lacks a well-motivated intro with more relevant references (currently only four).
There needs to be a description of the distinction between technological system shifts and socio-technical system shifts.
Some but not all key statements in 2.1 are connected to technological transitions.
The key lessons in 4.3 would benefit from being even more crisp and consider also the innovative processes within businesses generating the technological solutions, as mentioned under item 4 above.
Micro- and macroevolution are grouped, while examples of evolutionary principles related to microevolution are straightforward (flowers), the example on macro level is less intuitive to relate to.
Author Response
General Comments
Comments and Suggestions for Authors
This manuscript explores analogies between biological evolution and technological system transitions, aiming to derive insights for sustainability assessment. It is an ambitious and well-written manuscript, conceptually rich, and the result of substantial work.
I appreciate the authors' effort to initiate such a novel cross-disciplinary dialogue. However, I believe the manuscript, while promising, currently falls short of the depth and coherence needed for publication in its present form. My suggestions below are offered constructively to support further development of what could become a valuable contribution.
Strengths
Ambitious and interdisciplinary—addresses a compelling topic with relevance for biomimetics, transition studies, and sustainability metrics.
Strong conceptual and visual presentation (e.g., use of MLP, path dependency, glossary).
The analogy between evolutionary principles and transition mechanisms is intriguing and thought-provoking.
Answer: We would like to thank Reviewer 2 for her/his thorough review of the manuscript and suggestions, which substantially contributed to the improvement of our article. Our detailed answers are given in the Author's Notes. Any revision made to the manuscript are marked up in red. Please note that we have removed Figure 1 from the manuscript to shorten Section 2.1. In addition, we would like to point out that the Figures 4 and 6 (Please note: new numbering) have now been supplemented with public domain silhouettes for copyright reasons and thus differ slightly from the first versions. The figure captions have also been adapted accordingly. At the request of Reviewer 1, we present our main contributions comparing biological evolution and technological transition at the end of the Section “Introduction” (see page 4, 4th paragraph, lines 168-178).
Areas for Improvement
Comment-R2-01: Conceptual Focus and Scope -- The manuscript spans evolutionary biology, sustainability theory, system transitions, and implicitly philosophical considerations. While each theme is well-developed, the paper as a whole becomes diffuse and difficult to follow.
Suggestion: Sharpen the conceptual focus. Consider narrowing the scope based on key points you want to contribute to and clarify the intended audience.
Answer-R2-01: Given the interdisciplinary nature of the project, we agree with Reviewer 2 that we need to cover a wide range of topics. From other interdisciplinary projects we have experienced that we need a longer introduction in publications/presentations to provide all readers/audience with the necessary background information before we can present our new findings and results.
Also in the case of our manuscript, we cannot assume that all readers have substantial prior knowledge of biological evolution and the energy transition in the context of sustainability. For this reason, we had to provide more detailed information in some places to help readers understand the results (e.g., path dependency and MLP). For a better readability, we have shortened Section 2.1 by canceling some additional information in the text (see page 5, 1st paragraph, lines 194f.; page 5, 2nd paragraph, lines 211f.; page 5, 1st paragraph, lines 222-224) and by removing Figure 1 (see also Answer-R2-05). Additionally, we present our main contributions comparing biological evolution and technological transition at the end of the Section “Introduction”, which is certainly helpful for organizing a conceptual paper (see page 4, 4th paragraph, lines 168-178).
In fact, the manuscript has potential for additional analysis from a philosophical perspective (see also Answer-R2-02). Therefore, in future work, we are planning to include philosophers into our project team to consider the philosophical aspects in more detail. Thanks for the hint.
Comment-R2-02: Philosophical Assumptions about Evolution -- The paper treats evolution as purely random, however, this framing oversimplifies ongoing debates in evolutionary theory, especially concerning macroevolution, purpose, and emergence. Since sustainability is inherently teleological, this ontological tension needs addressing.
Suggestion: Acknowledge the limitations of the naturalistic framework and briefly reflect on alternative views (e.g., systems thinking, ontological pluralism).
Answer-R2-02: We would first like to state that we only included aspects of biological evolution that were necessary for the project's topic. Any further explanations would have exceeded the scope of the manuscript. We carefully selected the background information presented. In the following, we would like to comment individually on the various points raised:
- Randomness of biological evolution: Obviously, there has been a misunderstanding about the randomness of biological evolution. From a biological point of view, in addition to purely random events (e.g., mutation, recombination), natural selection is at work, a mechanism that is NOT random. Therefore we wrote in section 2.1.5: “Although frequently misinterpreted in the past, natural selection is considered to be the real controlling element of evolution since it compensates for the initial random processes of mutation and recombination, and determines the direction in which the gene pool of a population changes.” It seems that, regrettably, this central sentence was overlooked by the reviewer.
- Macroevolution: Concerning the debate on microavolution and macroevolution, we refer to Answer-R2-11.
- Purpose of biological evolution: From the point of view of evolutionary biologists, the beauty of biological evolution lies precisely in the fact that it has no inherent purpose, but that the entirety of recent and fossil species of the plant and animal kingdoms have evolved by trial and error (see Section 4.3). Thus, evolutionary biology does not have a purpose in the philosophical sense. Instead, all technologies have a purpose. To highlight this difference, we provided the relevant information in Section 2.1 (see page 4, 5th paragraph, line 182) and now added the word “purpose” to Section 4.3 (see page 22, 2nd paragraph, line 812).
- Emergence in biological evolution: To our knowledge, emergent evolution is the hypothesis that, in the course of evolution, some entirely new properties, such as mind and consciousness, appear at certain critical points, usually because of an unpredictable rearrangement of the already existing entities. Emergence is a fascinating topic that we have already covered in a separate publication (T. Speck & O. Speck, 2019, Emergence in biomimetic materials systems). However, we think that it would go beyond the scope of the manuscript if we were to include this aspect in the main body of the manuscript.
- Theory vs. mission statement: Since this is an important argument for us, we have addressed the tension between biological evolution and sustainability in detail in Section 4.3 “Theory vs. mission statement” and in the Discussion Section.
Suggestions: Against the background pointed out above, the suggestion goes far beyond the scope of our paper and the research questions in particular. We have now included the work of Arthur and Solée et al. in the introduction of the publication (see page 3, 5th paragraph, lines 134-140; page 4, 1st paragraph, lines 141-145) and compared their statement in the discussion section with our results (see page 25, 2nd paragraph, lines 975-996). However, we were unable to include further comments on alternative perspectives, such as systems thinking and ontological pluralism, in order to remain within the scope of this publication and to avoid making the publication even longer.
Comment-R2-03: Lack of Empirical Grounding -- Despite frequent mention of metrics and indicators, the paper remains purely conceptual. There is no application of the proposed ideas to a case study, nor any operational examples.
Suggestion: Could any minor worked example be included? Or discussion on how the presented Evolution Strategy. This would aid in demonstrating how the insights translate into actionable assessment tools.
Answer-R2-03: Thank you for raising the important issues of empirical underpinning and practical implementation of the results. In this context, we would like to emphasize that our primary purpose in writing the manuscript was to carefully develop the necessary foundations and limitations for comparing biological evolution and technological transition. Consequently, the manuscript is intended and designed as a conceptual paper first and foremost. The derived metrics and indicators can therefore only serve as a starting point for further development and refinement in order to become fully operational. Nevertheless, we have demonstrated how a start can be made with the example of the ASCR indicator presented in Section 4.4. With the amendments made in this section in response to the following comment, we have further clarified how the ASCR indicator can be implemented at the product development level and as part of R&D cycles (see page 24, 4th paragraph, lines 934-938). Concerning the translation of the results into actionable assessment tools for strategic decision making, we have pointed out that the new indicators will be integrated into TAPAS. TAPAS is intended as a strategic planning tool in the innovation process and is therefore ideally suited for implementing the indicators in the decision-making processes of ongoing research and development projects carried out by businesses or research institutions. Following your comment, we have highlighted these relationships more clearly in the Outlook (see page 27, 3rd paragraph, lines 1091-1094). The case studies and operational examples you have suggested must certainly be the next step, but they would clearly go beyond the scope of this manuscript and should therefore be reserved for a future publication.
Concerning Evolution Strategy: As outlined in Sections 2.2 and 4.3, Evolution Strategy is highly effective and robust in addressing maximization or minimization problems. We have added a case study showing how renewable electricity generation can be effectively placed and shared with existing ones respecting the constraints of the electrical grid (Dujardin et al. 2021, see page 11, 5th paragraph, lines 358-362). Another real-world example is evolution management, which borrows management strategies from the principles of biological evolution for the organization of companies (Otto & Speck 2011) and economical aspects of projects (Day et al. 2021). Since organizational management is primarily offered as a service by management consultancies and does not take a scientific approach, we only briefly touched on this topic for the sake of completeness in the discussion section (see page 25, 4th paragraph, lines 1016-1019; page 26, 1st paragraph, lines 1020-1023).
Comment-R2-04: Missing Innovation Dynamics within Firms -- While the paper draws system-level analogies between evolution and technological system change, it misses the rich evolutionary dynamics occurring within product development and R&D cycles—e.g., variation, selection, and abandonment of ideas.
Suggestion: Strengthen the analogy by incorporating more levels of innovative processes than on the macro transition level.
Answer-R2-04: Many thanks to Reviewer 2 for pointing out this gap. In Section 4.4 we have developed the ASCR indicator, which actually is intended to be implemented at the level of product development and as part of R&D cycles. Accordingly, we have made these relationships more explicit at the end of Section 4.4 (see page 24, 4th paragraph, lines 934-938).
Comment-R2-05: Readability and Length -- The manuscript is long and concept-heavy. While the glossary and structure help, the core contribution gets somewhat diluted.
Suggestion: Consider reducing length by summarizing background sections and focusing on the novel contributions and implications.
Answer-R2-05:
- Readability: As a conceptual paper, it needs to be profound and has to provide the description of the background and concepts. We have reorganized the introduction section by adding the main contributions, which should help to better structure our conceptional paper (see page 4, 4th paragraph, lines 168-178). In addition, we have tried to make the organization of the paper better understandable to the reader by using meaningful headings.
- Length: For a better readability, we have shortened Section 2.1 by canceling some additional information in the text and by removing Figure 1. The further explanations in Section 2.1 are necessary to understand the analogies between biological evolution and technical transformation in Sections 4.1 and 4.2 regarding path dependency and MLP. We have tried to meet the three reviewers' requests for changes without making the manuscript unnecessarily longer.
Final Recommendation
Comment-R2-06: I recommend a major revision. The manuscript offers valuable ideas, but it needs clearer framing and stronger philosophical grounding. I would like to thank the authors for an interesting read, which I believe will be improved through valuing the above-mentioned aspects and how these can improve the manuscript.
Answer-R2-06: We thank reviewer 2 for her/his opinion that our paper is an interesting read. We are confided that we significantly have improved the paper by following most of the general and specific suggestions of all reviewers. Unfortunately, we could not include all of the reviewer's suggestions in the manuscript. Due to the chosen scope of the paper, we were only able to briefly touch on the philosophical perspective. However, we have taken many new ideas into consideration for future projects.
Some specific comments:
Comment-R2-07: The introduction lacks a well-motivated intro with more relevant references (currently only four).
Answer-R2-07: We thank Reviewer 2 for the valuable feedback. Based on the recommendations of all three reviewers, we have reorganized the introduction. In the revised introduction, we cite and refer to 28 related papers. We gladly included the findings of additional relevant studies at the interface of biological evolution and technological transition, such as information about “combinatorial evolution,” a term coined by economist and complexity theorist W. Brian Arthur, as well as a study by Solée et al., who adopted Arthur's view that technological innovations depend heavily on the combination of pre-existing elements. This notion is nicely illustrated by the evolutionary tree of cornets. Additionally, we revised the structure of the introduction to make the motivation for our analyses more apparent. Along with the motivation specified by two key questions, we present our main contributions at the end of the introduction (see page 4, 4th paragraph, lines 168-178).
Comment-R2-08: There needs to be a description of the distinction between technological system shifts and socio-technical system shifts.
Answer-R2-08: Thank you for highlighting the importance of distinguishing between technological system shifts and socio-technical system shifts. We have included this distinction in Section 3.1, prior to the introduction of the MLP framework (see page 12, 3rd paragraph, lines 377-387). This addition will bring more clarity to the readers regarding the discussion.
Comment-R2-09: Some but not all key statements in 2.1 are connected to technological transitions.
Answer-R2-09: We have proved this recommendation thoroughly. From our point of view all information given in Section 2.1., except for the deleted sentence parts and the removed Figure 1, are necessary to understand the results presented in Section 4. For example, the explanations in Section 2.1 are essential for understanding Figure 7, which illustrates the role of mutation, recombination, gene drift, isolation, and natural selection. Without these explanations, transferring the theory of biological evolution to the path-dependency and lock-in model would be impossible, a transfer which, to our knowledge, is a completely new illustration.
Comment-R2-10: The key lessons in 4.3 would benefit from being even more crisp and consider also the innovative processes within businesses generating the technological solutions, as mentioned under item 4 above.
Answer-R2-10: Honestly, we are surprised that the presentation of our results in Section 4.3 do not seem “crisp” enough. We assure Reviewer 2 that, throughout the project, we tested various presentation formats to illustrate the similarities and differences between biological evolution and technological transitions. All of the authors agree that the current presentation is by far the best because the bold headings provide the reader with a quick overview. If they want more information, they can study the more detailed text of the respective bullet points. The problem may lie in the fact that we abbreviated the differences with “vs.,” which could easily be overlooked. To avoid any misunderstandings, we have changed all “vs.” to “versus” (see page 21, 6th paragraph, line 787; page 21, 7th paragraph, line 798; page 22, 2nd paragraph, line 808; page 22, 3rd paragraph, line 822).
Concerning innovative processes within businesses generating the technological solutions: As already mentioned in our response to Comment-R2-04, we have developed the ASCR indicator in Section 4.4, which is actually designed to provide guidance for innovation processes within businesses and research organizations generating the technological solutions. Accordingly, we have made these relationships more explicit at the end of Section 4.4 (see page 24, 4th paragraph, lines 934-938).
Comment-R2-11: Micro- and macroevolution are grouped, while examples of evolutionary principles related to microevolution are straightforward (flowers), the example on macro level is less intuitive to relate to.
Answer-R2-11: Thank you for pointing out the distinction between microevolution and macroevolution. We have discussed this point thoroughly and have concluded that introducing these terms is not helpful to the approach we are aiming for in our manuscript. In biology, the use of the terms “microevolution” and “macroevolution” is controversial. Many evolutionary biologists today avoid both terms, arguing that they are based on the same facts: “macroevolutionary” processes are merely a temporal summation of “microevolutionary” processes, making the distinction artificial and blurred. In order not to increase the length of the manuscript any further, we therefore would like to completely omit the distinction between micro- and macroevolution and the controversial discussion associated with it. However, we would like to point out that evolution is mostly a summation of microevolutionary processes over an evolutionary time scale, and have therefore included this point in the first sentence of Section 2.1 by adding “...which accumulate over generations on an evolutionary time scale” (see page 4, 5th paragraph, line 182).
Reviewer 3 Report
Comments and Suggestions for Authors
This research is generally well-written. I just have a few comments:
1) The authors claimed the similarity between biological evolution and technological transition. To strengthen this argument, more real-world examples should be provided.
2) The main research theme (i.e., what we can learn from biology) has been sufficiently discussed. However, this study would benefit from more detailed suggestions on how these insights could be applied to policy or strategy.
Author Response
General comments
Comment-R3-01: This research is generally well-written. I just have a few comments.
Answer-R3-01: We thank Reviewer 3 for taking the time to read and rate our manuscript. We have carefully gone through the comments and requests and found them all very valuable for further improving our manuscript. Our detailed answers are given in the Author's Notes. Any revision made to the manuscript are marked up in red. In addition, we would like to point out that the figures 4 and 6 (Please note: new numbering) have now been supplemented with public domain silhouettes for copyright reasons and thus differ slightly from the first versions. The figure captions have also been adapted accordingly. At the request of Reviewer 1, we present our main contributions comparing biological evolution and technological transition at the end of the Section “Introduction” (see page 4, 4th paragraph, lines 168-178).
Comment-R3-02: 1) The authors claimed the similarity between biological evolution and technological transition. To strengthen this argument, more real-world examples should be provided.
Answer-R3-02: With thank reviewer 3 for pointing out this interesting aspect of real-world examples. In the revised version, we provide a meaningful example using Evolution Strategy, which is highly effective and robust in addressing maximization or minimization problems. We have added a case study showing that Evolution Strategy can solve the problem how renewable electricity generation can be effectively placed and shared with existing ones respecting the constraints of the electrical grid (Dujardin et al. 2021, see page 11, 5th paragraph, lines 358-362). Another real-world example is evolution management, which borrows management strategies from the principles of biological evolution for the organization of companies (Otto & Speck 2011) and economical aspects of projects (Day et al. 2021). Since organizational management is primarily offered as a service by management consultancies and does not take a scientific approach, we only briefly touched on this topic for the sake of completeness in the discussion section (see page 25, 4th paragraph, lines 1016-1019; page 26, 1st paragraph, lines 1020-1023).
Comment-R3-03: 2) The main research theme (i.e., what we can learn from biology) has been sufficiently discussed. However, this study would benefit from more detailed suggestions on how these insights could be applied to policy or strategy.
Answer-R3-03: We thank Reviewer 3 for making this valuable recommendation. With regard to the practical application of the results into strategic decision making, we have put further emphasis on the integration of the new indicators into TAPAS in the Outlook section (see page 27, 3rd paragraph, lines 1091-1094). TAPAS is intended as a strategic planning tool in the innovation process and is therefore ideally suited for implementing the indicators in the decision-making processes of ongoing research and development projects carried out by businesses or research institutions. In terms of policy application, we have already discussed the interaction between technological niche innovations such as balcony power plants and flanking measures at the regime level at the end of Section 5.
Round 2
Reviewer 1 Report
Comments and Suggestions for Authors
I suggest accepting this article!
Reviewer 2 Report
Comments and Suggestions for Authors
Thank you for the revised manuscript. The implemented changes are creating a better presentation which can be considered sufficient for being published. Also, thanks for your thoughtful reflections and responses to my issues with the manuscript.
It would be interesting to follow the development of the philosophical dimensions in future work, especially when touching on the potential of purpose in biological evolution, as it is not pure randomness but an inherent mechanism of natural selection and fitness, which to me, as a systems thinker, on an ontological stance, may introduce thoughts on a more layered understanding of directionality in evolutionary processes (interplay of adaptation, fitness landscapes, and systemic constraints) that may point not to randomness alone, but to emergent patterns that resemble purpose or causality.